# Anisotropic Weighted Total Variation Feature Fusion Network for Remote Sensing Image Denoising

**Huiqing Qi** [1,2,3], **Shengli Tan** [1] **and Zhichao Li** [3,*]

1 School of Mathematical Sciences, East China Normal University, Shanghai 200241, China
2 Shanghai Institute of AI for Education, East China Normal University, Shanghai 200062, China
3 Key Laboratory of Land Surface Pattern and Simulation, Institute of Geographic Sciences and Natural Resources Research, Chinese Academy of Sciences, Beijing 100101, China
* Correspondence: lizc@igsnrr.ac.cn

**Abstract:** Remote sensing images are widely applied in instance segmentation and objetive recognition; however, they often suffer from noise, influencing the performance of subsequent applications. Previous image denoising works have only obtained restored images without preserving detailed texture. To address this issue, we proposed a novel model for remote sensing image denoising, called the anisotropic weighted total variation feature fusion network (AWTV$F^2$Net), consisting of four novel modules (WTV-Net, SOSB, AuEncoder, and FB). AWTV$F^2$Net combines traditional total variation with a deep neural network, improving the denoising ability of the proposed approach. Our proposed method is evaluated by PSNR and SSIM metrics on three benchmark datasets (NWPU, PatternNet, UCL), and the experimental results show that AWTV$F^2$Net can obtain 0.12~19.39 dB/0.0237~0.5362 higher on PSNR/SSIM values in the Gaussian noise removal and mixed noise removal tasks than State-of-The-Art (SoTA) algorithms. Meanwhile, our model can preserve more detailed texture features. The SSEQ, BLIINDS-II, and BRISQUE values of AWTV$F^2$Net on the three real-world datasets (AVRIS Indian Pines, ROSIS University of Pavia, HYDICE Urban) are 3.94~12.92 higher, 8.33~27.5 higher, and 2.2~5.55 lower than those of the compared methods, respectively. The proposed framework can guide subsequent remote sensing image applications, regarding the pre-processing of input images.

**Keywords:** remote sensing image; noise removal; weighted total variation; feature fusion; convolutional neural network

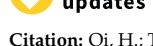



## 1. Introduction

Remote sensing imagery has been widely used in various fields, including land use and land cover mapping [1–3], forest fire recognition [4], military target reconnaissance [5], and so on. However, due to the irreducible influences of imaging equipment and the processes of image compression, transmission, and storage, the collected images will be distorted by random noise [6]. Image denoising is an ill-posed problem that aims to restore clean images, which can satisfy the visual perception of humans. The noise in collected remote sensing images can be divided into periodic and random, according to its manifestation [6]. We can improve the hardware equipment or model the periodic noise to eliminate periodic noise, but we cannot remove random noise by improving the imaging equipment, due to the fixed nature of the imaging system. According to the description in [7], we can divide random noise into multiplicative and additive noise, based on the relationship between the noise and image signals. The noise model can be written as the following inverse problem:

$$y_i = \phi(x_i) + N(0, \sigma^2), \tag{1}$$

where $y_i$ denotes the collected remote sensing image value at pixel $i$, $x_i$ represents the clean image value, the functional $\phi$ represents the noise which is multiplicative noise, $N(0, \sigma^2)$

denotes the signal-independent noise (i.e., additive Gaussian noise), and $\sigma^2$ is the parameter of the Gaussian distribution.

Researchers tend to use image-processing methods to eliminate remote sensing noise signals, and traditional image denoising methods are usually limited by prior knowledge about the noise [8] and the parameters of these algorithms must be tuned [9], which is inevitable and complex. Recently, various studies have proposed deep convolutional neural networks for image denoising. For example, Wang et al. [10] proposed a 3D convolutional neural network (CNN) for remote sensing image denoising; Feng et al. [11] utilized generative network (GN) technology to denoise remote sensing images; Dou et al. [12] used a spatial and spectral channel attention network for hyperspectral remote sensing image denoising. However, as remote sensing images contain rich texture information, previous deep neural networks could not allow the restored remote sensing images to preserve significant image textures efficiently. For this purpose, we proposed an anisotropic weighted total variation feature fusion network (AWTV$F^2$Net) for remote sensing image denoising. The proposed network consists of an anisotropic weighted total variation (AWTV-Net) module and four strengthen–operate–subtract boosting strategy modules (SOSB), three auto-encoder modules (AuEncoder), and three fusion block modules (FB). The image-denoising workflow of our model is shown in Figure 1.

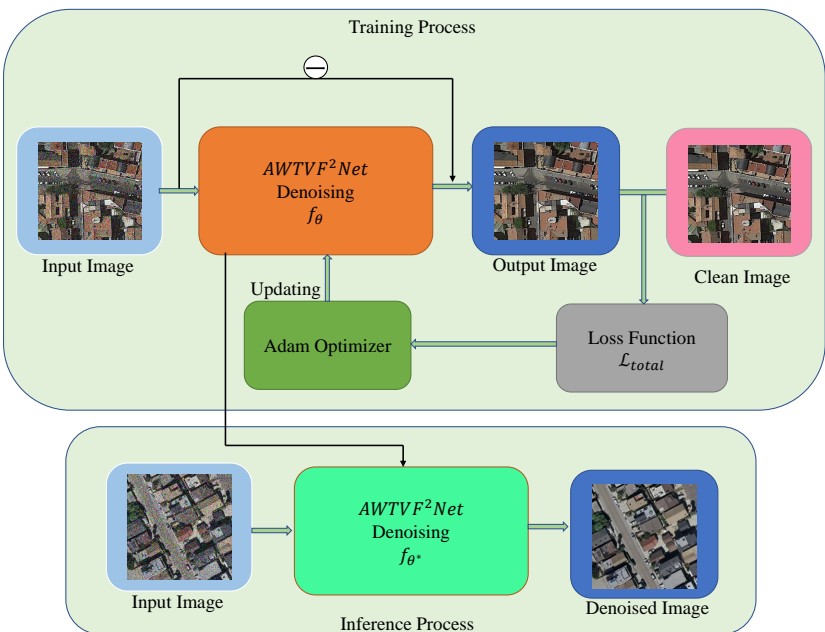

**Figure 1.** The workflow of our model for remote sensing image denoising. The first subfigure is the training process, which consists of the proposed network, optimizer, and loss functions. The second subfigure is the inference process, which uses the trained networks to restore remote sensing images.

In summary, our main contributions are as follows:

1.  To preserve the rich and complex texture features in remote sensing images, inspired by the anisotropic diffusion coefficient function, we proposed a new diffusion coefficient as a weighted function integrated into the traditional total variation (TV) model, and designed an AWTV-Net module that combines the weighted TV model into a deep learning method. The AWTV-Net module helps to extract rich features and detailed textures from noisy remote sensing images, strengthening the ability of our proposed model to preserve details.
2.  For multi-scale features, we design an AuEncoder module network architecture to extract multi-scale features from various input sources, consisting of maps transferred from the AWTV-Net module and the original noisy image. The former features are boosted by the proposed SOSB modules, providing detailed contents and texture

information, while the latter features provide the necessary information. The two kinds of features facilitate the subsequent noise reconstruction. As a result, we propose the FB module to combine the two above-mentioned types of multi-level feature maps. Notably, the fusion features improve the expressiveness of the proposed model. The use of multi-level feature fusion enhances the denoising ability while preserving details.

3.　The proposed network uses two weighted losses—$L_2$ loss and TV loss—which allow the network to converge better without losing image detail information. We conducted ablation experiments to confirm the weighting parameters of two losses. The optimal chosen parameters enable improvement of the noise removal capability.

4.　Extensive experiments on synthetic noisy remote sensing images and real noisy remote sensing images were conducted, in order to verify the effectiveness of the proposed AWTV$F^2$Net. Numerical metrics and visual comparisons indicate that our proposed anisotropic weighted total variation feature fusion network has significantly superior performance over the State-of-The-Art (SoTA) algorithms for remote sensing image noise removal.

The remainder of this article is organized as follows: In Section 2, we mainly describe related works focused on remote sensing image denoising and the motivation of our proposed model. Section 3 introduces the proposed method, which includes the proposed anisotropic weighted function and the details of the proposed network. We present the neural network settings and our method's implementation process in Section 4. Section 5 provides the comparative results of our comprehensive experiments, illustrating the competitive effectiveness of the proposed AWTV$F^2$Net. Section 6 discusses the limitations of this study and the differences among all the compared methods. Finally, our conclusions and directions for future research are given in Section 7.

## 2. Related Works

### 2.1. Traditional Methods of Remote Sensing Image Denoising

A growing number of approaches has been proposed and applied in remote sensing image denoising [13]. Traditional image noise removal methods can be divided into three categories: filter-based methods, total variation, and tensor decomposition-based algorithms.

Filter-based methods use the local information of the center pixel and remove noise according to the numerical relationship between the current pixel and neighboring pixels, such as mean filter [14], median filter [15], and Gaussian filter [16]. Filter-based methods tend to apply to image de-noising with a quick implementation property. However, filter-based methods provide the same coefficient in all directions, causing blurring in fine features. Buades et al. [17] have proposed a non-local mean method to overcome the limitations of filter-based methods. This method can obtain higher-quality restored images than local information-based filters. The block matching and three-dimensional filtering (BM3D) algorithm [18] is more complicated than the non-local mean method, which combines the advantages of the non-local mean method and wavelet transform domain methods.

The total variation model [9] was proposed in 1992, which aims to preserve image structure information. Total variation-based methods use regularization to constrain the denoising model, maintaining image texture features. Considering the spectral noise and spatial information difference in hyperspectral images, Yuan et al. [19] have proposed a spectral–spatial adaptive total variation (SSAHTV) model for hyperspectral remote sensing image denoising. Due to the structural sparsity of hyperspectral images, a group sparsity regularized hybrid spatial–spectral total variation (GHSSTV) [20] model has been proposed for hyperspectral image restoration. In recent years, deep image prior [21,22] information has been integrated into total variation models, which can make up for the unknown prior. However, the total variation-based methods require the tuning of complex parameters, which is usually not easy. Therefore, adaptive parameter selection for the weighted-TV model [23] has been used to address the complex parameters' tuning problem.

Tensor decomposition-based algorithms consider the inherent structure of remote sensing imagery, using low-rank tensor decomposition and recovery technology for hyperspectral image restoration. Wang et al. [24] have utilized the consistent structures between clean hyperspectral images and noisy images for image denoising by tensor decomposition and recovery. As the low-rank tensor approximation lost small content information, Zeng et al. [25] have added regularization to the low-rank tensor approximation item, which can improve the performance of remote sensing image restoration. Kong et al. [26] have proposed a framelet-tensor nuclear norm model for hyperspectral image denoising that takes full advantage of the redundancy of the framelet transform and the low-rank nature of the framelet-based transformed tensor. Overall, Tensor decomposition-based algorithms have excessively high computational and time costs, due to the need for decomposition on high-dimensional tensors.

In summary, the traditional methods have achieved remarkable performance in remote sensing image denoising. However, they have three key drawbacks: (1) it is hard to tune the various hyperparameters assigned manually for them; (2) they depend closely on the number of iterations and easily produce over-smoothing; and (3) they are typically only adaptive to a single type of noise.

### 2.2. Deep Learning Methods of Remote Sensing Image Denoising

Image noise removal technology based on deep learning methods has achieved remarkable results in recent years, which can be divided into supervised and unsupervised deep learning methods.

Supervised deep learning methods for image denoising involve the use of paired clean and noisy images to train neural networks. The multi-layer perception (MLP) denoising network [27], consisting of four fully connected layers, was proposed by Burger et al. It was the first time that deep learning achieved similar performance to that of the BM3D algorithm in the image restoration task. Considering the high spectral correlation between adjacent bands in hyperspectral images, Maffei et al. [28] have utilized a single supervised model for hyperspectral image denoising, obtaining better results than BM3D and the MLP-based model. Combining the local and global information of noisy remote sensing images, a deep spatial–spectral global reasoning network based on the U-Net architecture was invented by Cao et al. [29] in 2021. This model used the U-Net network to extract rich features from the input images, in order to obtain high-quality denoised images. Jia et al. [30] have proposed a dual-complementary convolution network (DCCNet) for remote sensing image denoising. The DCCNet uses a wavelet transform operation and combines it with a shuffling operation to recover the image structure and texture information. Ulyanov et al. [31] have used a generator network to learn the prior from the random input noise vector, then restored the image from the prior information by a decoding network. All of the models mentioned above are supervised methods and require clean images for supervised learning during network training.

Unsupervised deep learning involves training neural networks with noisy images, and does not require clean images as in surpervised learning. Unsupervised deep learning methods for image denoising are also called blind denoising methods. As clean remote sensing and medical images are hard to acquire, blind image denoising has become popular. There exist many blind denoising algorithms [32–37]. The "Noisy-As-Clean" (NAC) strategy [38] of training a self-supervised network for image denoising involves adding a simulated noise to the noisy image as input data and regarding the noisy images as target images. The NAC model further destroys the information of noisy images, leading to poor results. Huang et al. [39] have constructed paired images from the same noisy image using a random neighbor sub-sampler. This model avoids ruining the noisy image, but it lose information about the noisy image due to down-sampling, leading to unsatisfying results. A single-image capable speckling method for image denoising has been proposed by Wang et al. [40] in 2022, which presented better restoration results than previously mentioned unsupervised models.

The deep learning based image denoising methods also have drawbacks: (1) the extracted feature maps of these models are not rich and sufficient, and they only utilize a single kind of feature map extracted from networks; and (2) they only handled the specific noise and are not adaptive to mixed types of noise.

*2.3. Motivation*

The proposed model was inspired by the anisotropic diffusion coefficient function and total variation model. We first revisit the classic anisotropic diffusion coefficient-based method and total variation model to introduce our algorithm better.

2.3.1. Anisotropic Diffusion Coefficient Function

Many PDE-based methods deployed with various diffusion coefficient functions have been applied in the image restoration field. In 1990, Perona and Malik [41] proposed an anisotropic diffusion model with two different anisotropic diffusion coefficient functions. The PM model [41] is written as:

$$\begin{cases} I_t = \text{div}(c(x,y,t)\nabla I) & = c(x,y,t)\Delta I + \nabla c \cdot \nabla I, \\ I(x,y,0) = I_0, \end{cases} \tag{2}$$

where $I_0$ and $I$ are the noisy image and restoration image, respectively; and div, $\nabla, \Delta$ represent the divergence operator, gradient, and Laplacian operator with respect to the space variables, respectively. The above formula can be simplified into the isotropic heat diffusion equation when the coefficient function $c(x,y,t)$ is a constant. The diffusion coefficient $c$ is a function of the gradient magnitude, and also the range of coefficient $c$ is in $(0,1]$. In the PM model [41], two diffusion coefficient functions were proposed, written as follows:

$$f_1(||\nabla I||) = \exp\left(-(||\nabla I||/K)^2\right), \tag{3}$$

$$f_2(||\nabla I||) = \frac{1}{1 + (||\nabla I||/K)^2}, \tag{4}$$

where $I$ denotes the image, $K$ is a control parameter that keeps the fine and texture features from blurring, and the diffusion coefficient $f$ is a function of the gradient magnitude on the input image. It provides a small value in the large value of gradient magnitude while providing a large value in the small value of gradient magnitude. That coefficient function can smooth the image in homogeneous regions and stop diffusion on texture features. Usually, an effective diffusion coefficient function must satisfy the following requirements [41]:

1.  It must be a positive and strictly decreasing function of gradient magnitude;
2.  It must be a continuous and differentiable function;
3.  It must satisfy $\lim_{x\to+\infty} f(x) = 0, \lim_{x\to 0} f(x) = 1$.

In recent years, various diffusion coefficient functions have been derived from tangent sigmoid function proposed in [42], due to the better performance of Tansig function close to 0 faster than hyperbolic tangent [43] function. The tangent sigmoid function in [44] is written as:

$$tansig(x) = \frac{2}{1 + \exp(-2x)} - 1. \tag{5}$$

It is evident that the above *tansig* function is not a diffusion coefficient function, as it cannot satisfy the three criteria of the coefficient function. However, it is essential for designing a coefficient, and many coefficient functions have been inspired by it. An anisotropic diffusion coefficient function based on the tangent sigmoid function has been proposed in [43], which is written as follows:

$$f_3(x) = \frac{2}{1 + \exp(2abs(x/k)^2)}. \tag{6}$$

From Formula (6), this coefficient function can be regarded as a modified version of *tansig*; that is,

$$f_3(x) = 1 + tansig(-abs((x/k)^2)) = \frac{2}{1 + \exp{(2abs(x/k)^2)}}, \tag{7}$$

where $x$ denotes the gradient magnitude, $k$ is a constant value, and $abs(\cdot)$ represents the absolute operator. Obviously, $f_3$ satisfies the three properties of the diffusion coefficient. Therefore, $f_3$ can be used in the diffusion model. The curves of the tangent sigmoid function and $f_3$ are shown in Figure 2, where $K = 25$.

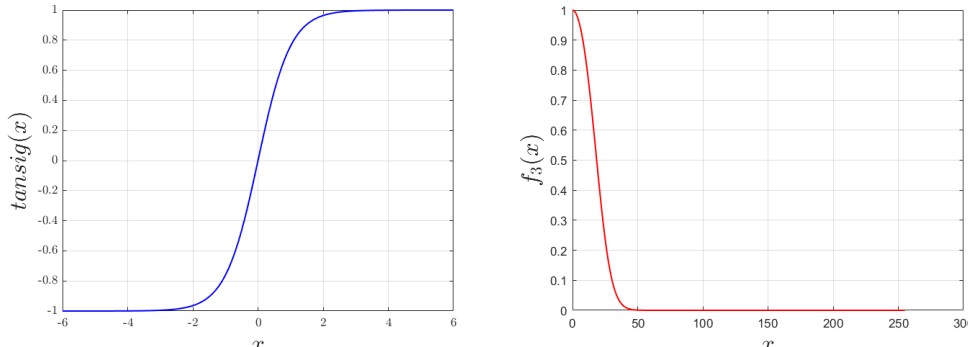

**Figure 2.** Curves of Tansig and $f_3$ coefficient functions.

From Figure 2, we can see that values of *tansig* function lie in $[0, 1]$ when $x \geq 0$. In contrast, the values of the diffusion coefficient function also lie in $[0, 1]$. This is why many diffusion coefficients are derived from the *tansig* function. The right picture is the curve of $f_3(x)$. According to the work of [41], if the value of the diffusion coefficient function is closer to 0, more small textures are retained in the denoised images, which are restored by the diffusion model. The same conclusion has been demonstrated in work related to $f_3(x)$. Our work was inspired by this conclusion, and we propose a novel anisotropic diffusion coefficient function, which is described in Section 3.

### 2.3.2. Total Variation Model

The total variation model is a classic regularized reconstruction-based method, which can convert the ill-posed image restoration problem into an optimization problem. The merit of the total variation model is that it can restore images while preserving edges, even when it was first proposed in [9]. The formula for total variation is written as follows:

$$\min_u \frac{1}{2} \sum_{i=1}^{n} (u - g)_i^2 + R(u), \tag{8}$$

where $u$ is the restored image and $g$ is the noisy image. The first term of above model is the fidelity term, while the second term is the regularization term. The fidelity always depends on noise with mean zero, and it is usually defined as a $l_2$ norm functional. For the regularization term, there exists a popular selection that is a total variation-based functional.

To solve the model of (8), we usually use the discrete form of $R(u)$ [9], defined as:

$$R(u) = \sum_{i=1]}^{n} ||(Du)_i||_p = \sum_{i=1}^{n} ((D_h u)_i^p + D_v u)_i^p)^{1/p}, \quad p \in \{1, 2\}, \tag{9}$$

where $(D_h u)_i, (D_v u)_i$ denote the horizontal and vertical discrete gradient of $u$ computed at pixel $i$, respectively; and $D_h, D_v$ are the first-order finite difference discrete operators. In this study, we only consider the case of $p = 2$. It is worth noting that there are many

related works considering $0 \leq p \leq 2$ in the total variation models. Therefore, according to (9), problem (8) can be rewritten as follows:

$$\arg\min_{u,t} \frac{1}{2} \sum_{i=1}^{n} (u - g)_i^2 + \alpha \sum_{i=1}^{n} ||t_i||_p, \tag{10}$$
$$\text{s.t.} \quad Du = t.$$

where the parameter $\alpha$ is used to control strength of the regularization term. Generally, we solve the minimization problem (10) using the augmented Lagrangian function in the ADMM framework [45], which has been widely applied to non-convex image restoration problems, as in [46].

In this research, we are greatly motivated by (10), and we replace $\alpha$ with an anisotropic diffusion coefficient, which differs in each pixel. Based on the above-mentioned anisotropic diffusion coefficients and basic total variation model, we propose a novel diffusion coefficient function and integrate the coefficient as a weight into the traditional total variation model, named the anisotropic weighted total variation model. The details of our proposed method and the solution to our model are described in Section 3 below.

## 3. Methodology

### 3.1. Anisotropic Weighted Total Variation

In this section, we mainly introduce our proposed model. Based on the description in Section 2.3, our model was greatly motivated by the anisotropic diffusion coefficient function and the traditional total variation model. We also start from the tansig function to derive our coefficient function. According to Equation (5), we have

$$\frac{1}{2} + \frac{1}{2} tansig(-(x/k)^2) = \frac{1}{1 + \exp\left(2(x/k)^2\right)}. \tag{11}$$

The above equation is close to the diffusion coefficient $f_3(x)$. Still, Formula (11) is not a diffusion coefficient, as it does not meet the three criteria of diffusion coefficient functions. The conclusion of [41] illustrated that the diffusion coefficient is faster close to 0, and the diffusion model can retain more details in restored images. Thus, we desired to design a novel coefficient that is the low bound of $f_3$ and satisfies the properties of diffusion coefficients. Inspired by above description of the anisotropic diffusion coefficient function, we proposed a new diffusion coefficient function, which is written as follows:

$$f_4(x) = \frac{1}{1 + x^2 \exp\left(2(x/k)^2\right)}, \tag{12}$$

Based on $f_3$, the proposed $f_4$ is the lower bound of $f_3$, which is demonstrated as follows:

$$1 + x^2 \exp\left(2(x/k)^2\right) \geq 1 + \exp\left(2(x/k)^2\right), \quad x \geq 1. \tag{13}$$

Then, we can obtain

$$\frac{1}{1 + x^2 \exp\left(2(x/k)^2\right)} < \frac{2}{1 + \exp\left(2(x/k)^2\right)}, \quad x \geq 1. \tag{14}$$

In this research, $x$ denotes the gradient value of pixels. We have $x \in \mathbb{Z}$. When $x = 0$, the left of the inequality (14) is equal to the right item. That is

$$\frac{1}{1 + x^2 \exp\left(2(x/k)^2\right)} = \frac{2}{1 + \exp\left(2(x/k)^2\right)}. \tag{15}$$

According to the Formulas (13)–(15), we have illustrated that $f_4(x)$ is the lower bound on $f_3(x)$ by mathematical theory. That also shows that our coefficient is better than $f_3(x)$.

It is worth noting that the proposed new diffusion coefficient function satisfies the three requirements mentioned in Section 2.3.1. $f_4$ is a continuous and differentiable function. As shown in Figure 3, $f_4$ is a positive and strictly decreasing function. Finally, $f_4$ also satisfies the limit condition, that is, $\lim_{x \to +\infty} f(x) = 0$, $\lim_{x \to 0} f(x) = 1$.

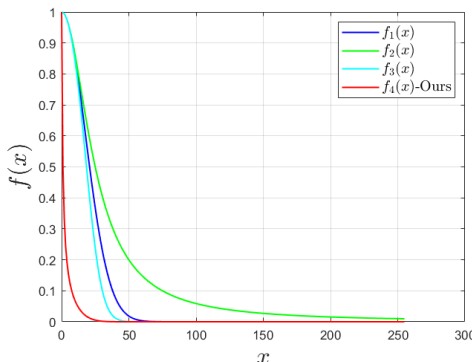

**Figure 3.** Comparisons of different diffusion coefficient function curves.

Figure 3 presents comparisons of our coefficient with other coefficient functions. Notably, our proposed coefficient approached zero faster than others, which means our coefficient can stop earlier at edges than others. Therefore, our coefficient can preserve more details of images. We introduced our novel anisotropic diffusion coefficient using mathematical theory and numerical simulation. The proposed coefficient function is better at retaining small contents than existing diffusion coefficients. Next, we introduce our modified anisotropic weighted total variation based on the proposed diffusion coefficient in the following.

Using a deep neural network to restore images involves a process that learns a map from noisy images to clean images. This process can be converted into a mathematical minimization problem, which reads as follows:

$$\theta^* \in \arg\min_{\theta} \mathcal{L}(f_\theta(g), u), \tag{16}$$

where $f_\theta$ denotes the proposed neural network with parameters $\theta$, $\mathcal{L}$ is the designed loss function (which we introduce in the following section), and $g$ and $u$ are paired noisy and clean images from a training set. Many methods have been proposed to solve (16), such as SGD and Adam. After we obtain $\theta^*$ by learning from the training dataset, we can infer $u^*$ from $g$; that is, $u^* = f_{\theta^*}(g)$.

Based on the above analysis, combining total variation in (10), we proposed the anisotropic weighted total variation model, written as:

$$\theta^* \in \arg\min_{\theta} \frac{1}{2} \sum_{i=1}^{n} (f_\theta(g) - g)_i^2 + \alpha \sum_{i}^{n} ||(Df_\theta)_i(g)||_p + \sum_{i}^{n} a_i ||(Df_\theta)_i(g)||_p,$$

$$a_i = f_4(||(Df_\theta)_i(g)||_p) = \frac{1}{1 + ||(Df_\theta)_i(g)||_p^2 \exp\left(2(||(Df_\theta)_i(g)||_p / k)^2\right)}, \tag{17}$$

To simplify the proposed model, (17) can be rewritten as follows:

$$\theta^* \in \arg\min_{\theta} \frac{1}{2} \sum_{i=1}^{n} (f_\theta(g) - g)_i^2 + \sum_{i}^{n} (\alpha I + a_i) ||(Df_\theta)_i(g)||_p, \tag{18}$$

where $I$ denotes the identity matrix, $D$ represents the discrete first-order finite difference operator as described in (9), and $||.||$ denotes the norm computation operator based on the

value of $p$, which can be assigned as 1 or 2; in this paper, we mainly discuss the model with $p = 2$. The constrained equivalent formula of (18) reads as:

$$\{\theta^*, t^*\} \leftarrow \arg\min_{\theta, t} \frac{1}{2} \sum_{i=1}^{n} (f_\theta(g) - g)_i^2 + \sum_i^n (\alpha I + a_i)||t_i||_p \quad p = 2,$$
$$\text{s.t.} \quad Df_\theta(g) = t. \tag{19}$$

We use the **ADMM** algorithm [45] to solve the non-convex problem of (19). Therefore, the augmented Lagrangian function of our model is defined as:

$$L(\theta, t, \lambda_t) = \frac{1}{2} \sum_{i=1}^{n} [(f_\theta(g) - g)_i^2 + 2(\alpha I + a_i)||t_i||_2 + \beta_t (Df_\theta(g) - t)_i^2] + <\lambda_t, Df_\theta(g) - t>, \tag{20}$$

where $\beta_t, \lambda_t$ denote the penalty parameter and the Lagrangian parameter, respectively. Here, $\beta_t$ must satisfy $\beta_t > 0$. In the **ADMM** framework [45], we find the solution of (20) by a minimization step for the primal variable $\theta, t$, and a maximization step for the dual variable $\lambda_t$. Considering the $k$th iteration of the **ADMM** algorithm [45], the three primal sub-problems of our model can be written as follows:

$$\theta^{k+1} \in \arg\min_\theta \frac{1}{2}||f_\theta(g) - g||_2^2 + \frac{\beta_t}{2}||Df_\theta(g) - t^k + \frac{\lambda_t^k}{\beta_t}||_2^2, \tag{21}$$

$$t^{k+1} = \arg\min_t \sum_{i=1}^{n} (\alpha I + a_i)^k ||t_i||_2 + \frac{\beta_t}{2}||t - Df_{\theta^{k+1}}(g) - \frac{\lambda_t^k}{\beta_t}||_2^2, \tag{22}$$

$$\lambda_t^{k+1} = \lambda_t^k + \beta_t (Df_{\theta^{k+1}}(g) - t^{k+1}). \tag{23}$$

The three primal sub-problems can be solved efficiently, in closed form, by shrinkage or projection operators and linear system solvers. In the second problem (22), the iterative form of the proposed weight $a_i$ is written as:

$$a_i^k = f_4(||(Df_{\theta^{k+1}})_i(g)||_2) = \frac{1}{1 + ||(Df_{\theta^{k+1}})_i(g)||_2^2 \exp\left(2(||(Df_{\theta^{k+1}})_i(g)||_2/k)^2\right)}. \tag{24}$$

The anisotropic weight regularizes more in the homogeneous region, and regularizes less in the areas with detailed complex texture.

### 3.2. Network Architecture

In this subsection, we mainly present the proposed network architecture and its components. Our network consists of four parts: The AWTV-Net module, four SOSB modules, three AuEncoder modules, and three FB modules. The detailed architecture of AWTV$F^2$Net is shown in Figure 4.

U-Net [47] has been widely used in different computer vision tasks, due to its strong ability to extract features. The proposed AWTV-Net module of our network is a modified version of U-net. The main difference between AWTV-Net and U-Net is the down-sampling operator. The implementation of the down-sampling operator in U-Net is max-pooling, while the down-sampling operator in AWTV-Net is a convolution layer with a stride of 2. The channel of intermediate output features is 128, except for the skip connection output. We set the skip size to 4. The skip connection output features are inputs to the SOSB module that can enrich feature space for the restoration of clean images. The final output of AWTV-Net is used to calculate the anisotropic weighted TV loss, which can preserve fine features in the denoised images. We enrich the feature space of model using the final output of AWTV-Net. Therefore, AWTV-Net provides multi-scale features by anisotropic weighted total variation, which prevents some crucial details from being smoothed in the restored images.

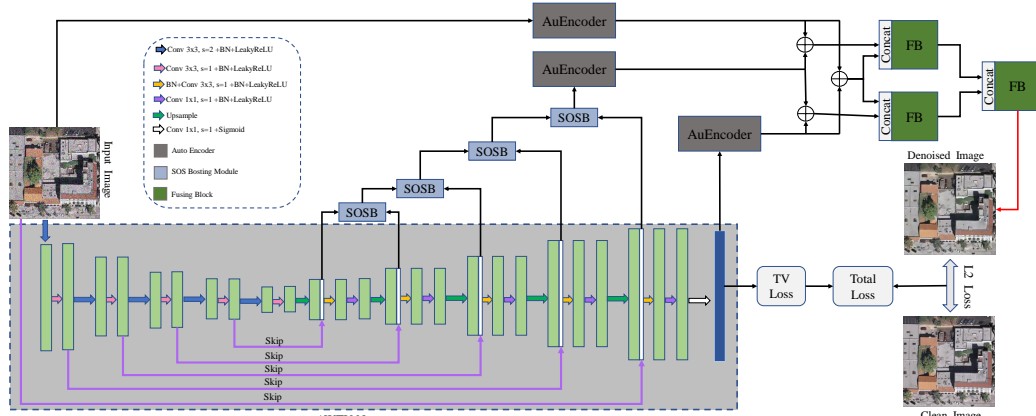

**Figure 4.** Network architecture of the proposed AWTV$F^2$Net. In this framework, the light green rectangles denote the middle features of the AWTV-Net module. The blue rectangle is the final output feature from AWTV-Net. The white rectangles are the skip feature maps. The thick, different-colored arrows indicate convolutional operators with different kernel sizes and strides, described in the middle dot-line rectangle. The black arrows denote the directions of data flowing. The SOSB rectangles are the feature map boosting modules, and AuEncoder rectangles are the modules extracting features from multi-inputs, $\oplus$ denote additive operator. FB rectangles indicate the feature fusion modules. The red arrow marks the process of image reconstruction. The loss rectangles represent different loss calculations.

The Strengthen–Operate–Subtract boosting (SOSB) strategy was first used in [48]. Its function is to operate the refinement process on the strengthened image, based on the previously estimated image in image-denoising tasks. In our network, we use the SOSB strategy to work on all outputs of skip connections. The mathematical formula of the SOSB strategy in [48] can be written as follows:

$$j^n = R_{\theta_n}^n\left(i^n + (i^{n-1}) \uparrow_2\right) - (i^{n-1}) \uparrow_2, \tag{25}$$

where $R_{\theta_n}^n$ is the refinement unit at the $n$th level parameterized by $\theta_n$, $\uparrow_2$ denotes an up-sampling operator with a scale parameter of 2, $(i^n + (i^{n-1}) \uparrow_2)$ is the strengthened feature, and $j^n$ is the inputs to the next layer (as $i^n$). In this work, each refinement unit is embedded into a residual group. Hence, the SOSB strategy [48] can be simplified as follows:

$$j^n = R_{\theta_n}^n\left(i^n + (i^{n-1}) \uparrow_2\right) + i^n. \tag{26}$$

We present the SOSB module architecture of our network in Figure 5. A deconvolution layer replaces the up-sampling operator in the SOSB module. The trainable refinement unit comprises a convolution layer, followed by a BatchNorm operator and a LeakyReLU function. The SOSB module is a bridge that connects the output maps of skip layers with the main framework of restored images. Meanwhile, it also enriches the feature space, which preserves some details on denoised images.

To better introduce the AuEncoder and FB modules, we first describe four important sub-architectures, as shown in Figure 6. Figure 6a,b show the EB (Extracting Block) and RB (Residual Block), respectively. Similar structures have also been used in [49], named as DB [50] and RB [51]. The proposed EB and RB are motivated by DB and RB, but with some differences. The extracting block includes two convolution operators, followed by a BatchNorm layer and a ReLU layer. At the end of EB, we concatenate the inputs and outputs of the module. The residual block differs from the original RB and that of [49]. We add a concatenate layer among the six layers in the RB. The first three layers are used to extract features, and the latter are for changing the dimension of images. Note that the proposed EBs possess powerful feature extraction capabilities, similar to DBs. Therefore,

we propose an EBS (Extracting Blocks Set) module that includes several extracting blocks, as shown in Figure 6c. The concatenate layer is also utilized in the EBS. The function of last three layers is to adjust the dimension of features. The proposed RBS (Residual Blocks Set) module has the same workflow as EBS. The RBS consists of several RBs. To solve the dimension change caused by the concatenation layer, we take the same approach as in EBS for adjusting the dimension of features at the end of RBS module. Both EBS and RBS possess powerful feature extraction capabilities. We use them to construct more powerful modules for extracting features and feature fusion. The rich maps extracted by EBS and RBS help to improve the model performance and allow more critical details to be preserved in the restored remote sensing images.

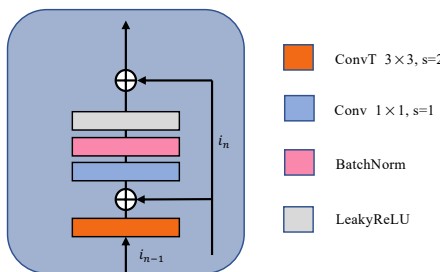

**Figure 5.** Intuition architecture of the SOSB module.

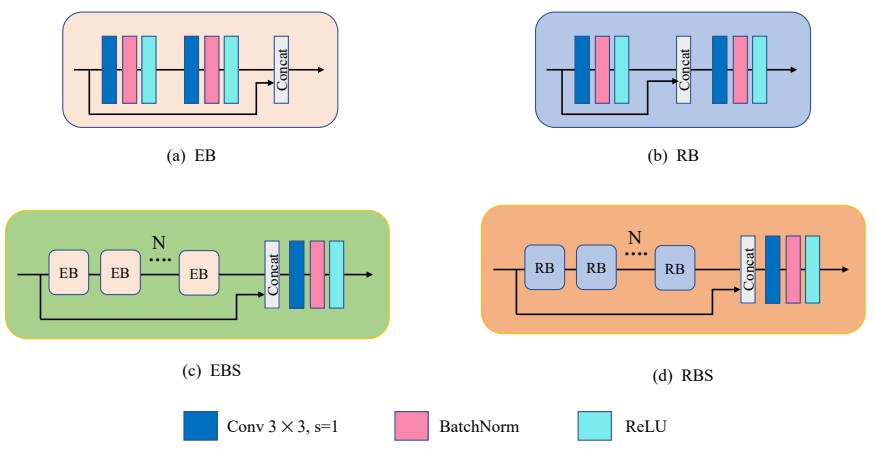

**Figure 6.** Architectures of (**a**) EB, (**b**) RB, (**c**) EBS, (**d**) RBS.

Based on the proposed EBS and RBS modules, AuEncoder and FB modules (shown in Figure 7) are introduced in the following. The AuEncoder (Auto-Encoder) module consists of an EBS, and an RBS. As the AuEncoder accepts multi-scale inputs, we add three layers at the head of AuEncoder module to make the dimension of the inputs uniform. The merit of AuEncoder is that it can extract rich features from different inputs, which is helpful for feature fusion in the downstream task. We also propose an FB (Fusing Block) module for feature fusion in the downstream task in this work. The FB module only consists of an RBS, which accepts multi-level features. To solve the problem caused by different dimensionalities, we adjust the dimensions of the multi-level features by a convolutional layer with a kernel size of 1. The convolution layer is followed by a BatchNorm layer and a ReLU layer. We demonstrate this process using mathematical formulas to understand the feature-fusion process efficiently.

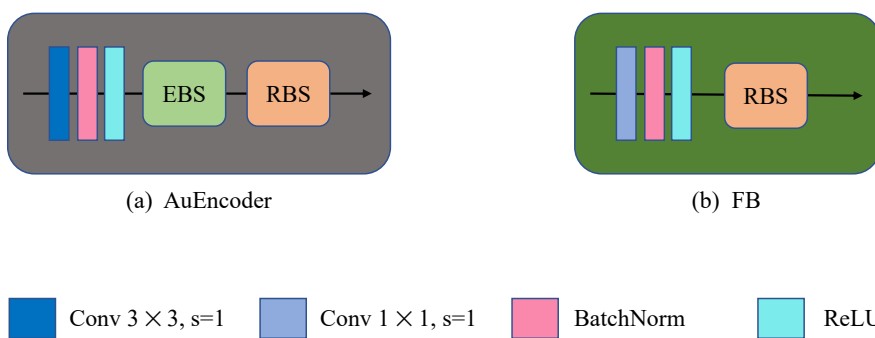

(a) AuEncoder　　　　　　　　　　　　　　　　　　(b) FB

| ■ Conv 3 × 3, s=1 | ■ Conv 1 × 1, s=1 | ■ BatchNorm | ■ ReLU |

**Figure 7.** Architectures of (**a**) AuEncoder, (**b**) FB.

As shown in Figure 4, we denote the features extracted by the three AuEncoders as $f_{noise}$, $f_{skip}$ and $f_A$; that is, $f_{noise}$ denotes features extracted by AuEncoder from noise images, $f_{skip}$ represents features extracted from outputs of SOSB modules, and $f_A$ denotes features from the final output of AWTV-Net. All $\oplus$ in Figure 4 can be written as follows:

$$f^1_{fuse} = f_{noise} + f_{skip}, \tag{27}$$

$$f^2_{fuse} = f_{noise} + f_A, \tag{28}$$

$$f^3_{fuse} = f_{skip} + f_A. \tag{29}$$

Then, we put the three $f^i_{fuse}$ features into the following two FB modules. The process can be written as follows:

$$I^1 = FB^1([f^1_{fuse}, f^2_{fuse}]), \tag{30}$$

$$I^2 = FB^2([f^2_{fuse}, f^3_{fuse}]), \tag{31}$$

where $[\cdot]$ denotes the concatenation operation. $I^i$ will be sent to the third FB module, which is close to the tail of the whole network:

$$I^3 = FB^3([I^1, I^2]), \tag{32}$$

where $I^3$ denotes the final output of our network. The red arrow in Figure 4 represents a convolution layer, used to restore the dimension of the denoised images. AWTV$F^2$Net has a powerful feature extraction capability, working on multi-scale feature maps. This nature of AWTV$F^2$Net preserves some critical details of the restored remote sensing images. AWTV$F^2$Net, as an end-to-end framework, is the first model that fuses the feature maps from the TV model.

*3.3. Loss Function*

According to our proposed network architecture, the use of fusion features helps to preserve complex textures. Our proposed model integrates anisotropic weighted total variation into the deep learning framework. Therefore, the loss function in our proposed AWTV$F^2$Net consists of the total variation loss and $l_2$ loss. The total variation loss function is defined as in (21), which reads as follows:

$$\mathcal{L}_1 = \frac{1}{2}||f_\theta(g) - g||_2^2 + \frac{\beta_t}{2}||Df_\theta(g) - t^k + \frac{\lambda_t^k}{\beta_t}||_2^2 \tag{33}$$

We utilize the mean square error function as the second part loss to measure the difference between denoised images restored by AWTV$F^2$Net and clean images. This part loss is written as:

$$\mathcal{L}_2 = \frac{1}{n}\sum_{i}^{n}(f_\theta(g) - x)_i^2, \tag{34}$$

where $x$ is the clean image, we have weighted the two parts of loss as the total loss by introducing parameters $\gamma_1$ and $\gamma_2$ in our algorithm. That reads as follows:

$$\mathcal{L} = \gamma_1 \mathcal{L}_1 + \gamma_2 \mathcal{L}_2, \quad \text{s.t.} \quad \gamma_1 + \gamma_2 = 1. \tag{35}$$

where $\gamma_1$ and $\gamma_2$ are two weights, which are positive. The two parameters are selected manually, and the effect of $\gamma_1$ and $\gamma_2$ on restored results are discussed in Section 5.

## 4. Experiments

In this section, we detail the used datasets, settings in the experiments, implementation details, environment settings of the used computer, methods for comparison, and evaluation metrics.

### 4.1. Datasets

#### 4.1.1. Benchmark Datasets

We used three benchmark datasets in our experiments to train and test AWTV$F^2$Net. These three datasets were NWPU-RESISC45 (NWPU) [52], PatternNet [53], and UC Merced land-use dataset (UCL) [54]. Northwestern Polytechnical University created the NWPU-RESISC45 dataset for remote sensing image scene classification. The NWPU-RESISC45 dataset includes 45 classes, where each class has 700 images with a size of 256 × 256. All of these images are colorful, with a spatial resolution from 0.2–30 m for most scene classes. The PatternNet dataset contains 38 classes, and each of which has 800 images. All remote sensing images in PatternNet have a spatial resolution from 0.062–4.693 m with a size of 256 × 256 in the RGB color space. The UC Merced land-use dataset is a freely and publicly available remote sensing image dataset, composed of 21 classes with a total number of 2100 images. The images in the UC Merced land-use dataset have a spatial resolution of 0.3 m and a size of 256 × 256 in the RGB color space.

#### 4.1.2. Real-World Datasets

We compared the performance between our model and other methods on three real-world datasets: AVRIS Indian Pines dataset [55], the ROSIS University of Pavia dataset [56], and the HYDICE Urban dataset [24]. The AVIRIS Indian Pines dataset was gathered by an AVIRIS sensor over the Indian Pines test site in North-western Indiana, and consists of 145 × 145 pixels images with 224 spectral reflectance bands in the wavelength range of 0.4~2.5×10$^{-6}$ m. The AVIRIS Indian Pines dataset includes some bands affected by the mixture of Gaussian and impulse noise, as introduced in [55]. For this study, we have used bands 3~6 to test our model. The ROSIS University of Pavia dataset is a hyperspectral image dataset gathered by a ROSIS sensor over Pavia, Italy. The size of images is 610 × 340 with 103 spectral bands. The geometric resolution of the images in the ROSIS University of Pavia dataset is 1.3 m. We used the first six bands' images in our experiments. The HYDICE Urban dataset is one of the most widely used hyperspectral remote sensing image datasets. The size of these images is 307 × 307, where each pixel corresponds to a 2 × 2 m$^2$ area. There are 210 wavelengths ranging from 400 nm to 2500 nm, resulting in a spectral resolution of 10 nm. We have chosen bands 138, 203, and 205 for our experiments.

### 4.2. Implementation Details

We used the NWPU-RESISC45 dataset to train our proposed model, and used the other two datasets to verify the performance of the new model. The NWPU-RESISC45 dataset was used to synthesize four different cases of noise for training the proposed model. The four cases of noise were described as follows:

(1) **Gaussian**: White Gaussian noise with zero mean and different noise levels from 15 to 50 ($\sigma^2$ = 15, 25, 35, and 50) was added to the training and testing images. For each noise level, we have trained and tested our model on all three datasets.

(2) **Gaussian + salt&pepper**: A mixture of Gaussian noise (zero mean, $\sigma^2 = 15$) and salt&pepper noise (with a parameter of 0.005) was used in our training and testing experiments.

(3) **Gaussian + speckle**: A mixture of Gaussian noise (zero mean, $\sigma^2 = 15$) and speckle noise (with a parameter of 0.005) was added to the training and testing datasets.

(4) **salt&pepper + speckle**: A mixture noise of salt&pepper and speckle noise (with a unified parameter value of 0.005) was added to all training and testing datasets.

The NWPU-RESISC45 dataset, with the above four different cases of noise added, was used to train our model. To verify and compare the performance of our model, we randomly selected 50 images from the NWPU-RESISC45 dataset and the PatternNet dataset, respectively, and 40 images were chosen from the UC Merced land-use dataset randomly for testing experiments. To all testing images, the same four cases of noise mentioned above were added.

The detailed architecture of our network is shows in Figure 4. Images of the NWPU-RESISC45 dataset were added with different types of noise for training. We empirically set $\alpha = 0.5$, $\beta_t = 10$ in (22), $\gamma_1 = 0.4$ and $\gamma_2 = 0.6$ in (35). There were some other hyper-parameters in our network, such as the number of RBs (assigned as 4), and EBs (set empirically with a value of 3). The parameters of the ReLU and LeakyReLU function were set as defeated (0.2). We set the skip size to a value of 4 in skip connection layers. The batch size was 4 for loading input data. The learning rate is initialized as $10^{-3}$ and decayed by 0.1 every 15 epochs. The total number of epochs was 30.

The proposed AWTV$F^2$Net was implemented using the PyTorch framework and parameters were updated using the Adam optimizer. The $\beta_1$, $\beta_2$ values of the Adam optimizer were 0.9 and 0.99, respectively. All experiments in this work are implemented in a computer server equipped with an E5-2698 CPU and 4 GTX 3090 AERO GPUs, where each GPU has 24 GB RAM. In addition, we used Python (version 3.6.13) and PyTorch (1.8.0) to conduct the training and testing experiments.

### 4.3. Model Comparison and Evaluation

In order to verify the performance and effectiveness of our proposed method, we compared AWTV$F^2$Net with several image denoising methods, including WNNM [8], BM3D [18], ADNet [57], DnCNN [58], ECNDNet [59], DIP-TV [60], and DIP-WTV [22]. These compared methods were tested on the three test datasets mentioned above. We not only compared the performance in terms of visual perception, but also evaluated the effectiveness of our model against the others through numerical metrics. Peak signal-to-noise ratio (PSNR) and structural similarity index measurement (SSIM) have been widely used to measure the quality of images restored from noisy images.

We used the PSNR and SSIM metrics to evaluate the performance of different methods. The PSNR is defined as follows:

$$\text{PSNR} = 10\log_{10}\left(\frac{b^2}{\text{MSE}}\right),$$
$$\text{MSE} = \frac{1}{M \times N} \sum_{x=0}^{M-1} \sum_{y=0}^{N-1} (I_c(x,y) - I_t(x,y))^2, \tag{36}$$

where $b$ is the maximum value of the gray level (generally, $b$ is assigned as 255 for 8-bit image). MSE is the mean square error evaluated from the clean and denoised images, $M \times N$ is the size of original image, and $I_c$, $I_t$ represent the clean image and intermediary image in the diffusion process, respectively. A high value PSNR value indicates a better-restored image. The second metrics, SSIM, is written as follows:

$$\text{SSIM} = \frac{(2\mu_x\mu_y + \varepsilon_1)(2\text{cov}(I_c, I_t) + \varepsilon_2)}{(\mu_x^2 + \mu_y^2 + \varepsilon_1)(\sigma_x^2 + \sigma_y^2 + \varepsilon_2)}, \tag{37}$$

where $\mu_x, \mu_y$ denote the average patch pixel values of the clean and denoised images, respectively; $\sigma_x, \sigma_y$ are the variances of patch pixel values for the clean image and denoised image, respectively; cov is covariance operator; and $\varepsilon_1, \varepsilon_2$ are two control parameters, which are used to stabilize the division with small denominators. The range of SSIM is $[0, 1]$, where a value closer to 1 indicates higher quality of the restored image. In this work, we mainly use the PSNR and SSIM metrics to compare performance of our proposed AWTV$F^2$Net with that of other methods.

## 5. Results

This section mainly describes the results of the comprehensive experiments conducted between the proposed method and compared methods on the considered datasets. This section consists of four components: The results for different noise levels (Gaussian), results for mixed noise, results of ablation experiments, and results of comparisons on real noisy remote sensing images, respectively.

### 5.1. Experimental Results of Additive White Gaussian Noise

To verify the performance of our proposed method, we present the results under different noise levels of white Gaussian noise, with $\sigma^2 = \{15, 25, 35, 50\}$, on the three different datasets. Figures 8–11 present the visual results of all compared methods. Table 1 provides quantitative numerical results for all methods, according to the PSNR and SSIM metrics.

**Table 1.** Comparison results of different methods with four noise levels on the three datasets. The bold values denote the best results.

| Datasets | Method | WNNM | BM3D | ADNet | DnCNN | ECDNet | DIPTV | DIPWTV | AWTV$F^2$Net |
|---|---|---|---|---|---|---|---|---|---|
| Metrics [1] | $\sigma^2$ | P/S | P/S | P/S | P/S | P/S | P/S | P/S | P/S |
| NWPU | 15 | 30.73/0.8856 | 31.28/0.9076 | 25.43/0.5812 | 24.54/0.5377 | 24.54/0.5388 | 28.31/0.8462 | 29.53/0.8752 | **32.27/0.9393** |
| | 25 | 28.01/0.8515 | 29.28/0.8757 | 24.46/0.5440 | 23.55/0.4834 | 23.54/0.4831 | 27.21/0.8212 | 27.52/0.8267 | **29.57/0.8975** |
| | 35 | 26.21/0.7500 | 28.21/0.8602 | 23.96/0.5080 | 22.73/0.4288 | -/- | 25.34/0.7498 | 24.64/0.7164 | **28.33/0.8663** |
| | 50 | 24.18/0.6600 | 26.05/0.7972 | 22.94/0.4529 | 21.68/0.3595 | 21.69/0.3616 | 22.89/0.6322 | 22.77/0.6314 | **27.00/0.8301** |
| PatternNet | 15 | 31.22/0.9014 | 31.48/0.9209 | 25.57/0.6379 | 24.87/0.5948 | 24.87/0.5961 | 27.95/0.8008 | 28.65/0.8081 | **32.48/0.9323** |
| | 25 | 28.56/0.8156 | 30.64/0.8993 | 24.98/0.6006 | 24.05/0.5431 | 24.04/0.5432 | 26.51/0.7550 | 26.34/0.7465 | **30.79/0.9017** |
| | 35 | 26.65/0.7516 | 28.55/0.8546 | 24.38/0.5636 | 23.29/0.4909 | -/- | 25.08/0.6988 | 24.76/0.6896 | **28.83/0.8578** |
| | 50 | 24.40/0.6710 | 26.14/0.7922 | 23.36/0.5052 | 22.21/0.4262 | 22.20/0.4277 | 22.86/0.5910 | 22.89/0.6038 | **27.90/0.8395** |
| UCL | 15 | 30.34/0.8843 | **31.02**/0.9152 | 25.02/0.5468 | 24.30/0.5057 | 24.30/0.5065 | 28.13/0.8389 | 28.31/0.8310 | 30.84/**0.9204** |
| | 25 | 28.64/0.8203 | 28.96/0.8561 | 24.44/0.5135 | 23.49/0.4556 | 23.48/0.4553 | 26.99/0.8093 | 26.28/0.7742 | **29.36/0.8852** |
| | 35 | 26.82/0.7601 | 27.49/0.8243 | 23.93/0.4835 | 22.81/0.4078 | -/- | 25.25/0.7388 | 23.79/0.6900 | **27.97/0.8420** |
| | 50 | 24.70/0.6797 | 26.43/0.8108 | 23.17/0.4359 | 21.93/0.3500 | 21.93/0.3516 | 22.29/0.6010 | 21.71/0.5919 | **27.19/0.8250** |

[1] P: PSNR, S: SSIM.

Figure 8 shows a visual comparison between proposed AWTV$F^2$Net and other four compared methods with a noise level of $\sigma^2 = 15$. We also present the PSNR and SSIM values under these sub-figures. It is notable that AWTV$F^2$Net obtained the highest PSNR and SSIM values, demonstrating that our proposed method has the best performance on this level of white Gaussian noise. Meanwhile, we can see, from Figure 8, that denoised images restored by DnCNN and ADNet methods were distorted, and that DnCNN and ADNet have obtained the lowest PSNR and SSIM values. WNNM and DIPWTV obtained competitive results, but the images restored by them were over-smoothed, with consequent loss of fine features and details.

We also tested the performance of the proposed AWTV$F^2$Net with white Gaussian noise at $\sigma^2 = 25$. The restored images, in comparison with other methods are shown in Figure 9. These results proved that our AWTV$F^2$Net method had the best performance among the compared methods—whether from visual comparison or numerical metrics. From Figure 9, it is evident that, in the images restored by the DIPWTV method for the three datasets, there exists a water ripple phenomenon in the images, thus reducing the quality of denoised images. Meanwhile, the images restored by DnCNN and ADNet were distorted, and the background colors had changed in these images. The WNNM method has restored better images than DIPWTV, DnCNN, and ADNet. Notably, our proposed

method has higher quality restored images than WNNM, as the proposed AWTV$F^2$Net retained more details in the denoised images. The highest values of PSNR and SSIM also illustrate that our method had the best performance among all compared methods.

Figure 10 shows denoised images on NWPU, PatternNet, and UCL datasets with additive white Gaussian noise $\sigma^2 = 35$. Meanwhile, we also compared our method with other methods for a higher noise level ($\sigma^2 = 50$), as shown in Figure 11. From Figure 10, we can also see water ripples on the denoised images obtained by the DIPWTV method. DnCNN and ADNet methods caused mild color distortion on the PatternNet dataset; although they had competitive quality in restored images on both PatternNet and UCL datasets, they caused serious color distortion on the UCL dataset. Compared with WNNM method, they could obtain better visually denoised images, but WNNM has smoothed fine features and details. In contrast, our method obtained the highest quality restored images, as well as PSNR and SSIM values. The same manifestations of all compared methods can be observed in Figure 11.

Particularly, in Figure 11, it is notable that ADNet and DnCNN presented weakened color distortion on those images with a dack background. However, they also presented a distortion effect on the images of the UCL datasets, leading to negative SSIM values (we use the symbol '-' to denote a negative SSIM value). The WNNM method has also smoothed the small image content in these results. The proposed AWTV$F^2$Net restored the denoised images with high quality and prevented small image content from being smoothed. Thus, our method retained more important fine features in these restored remote sensing images.

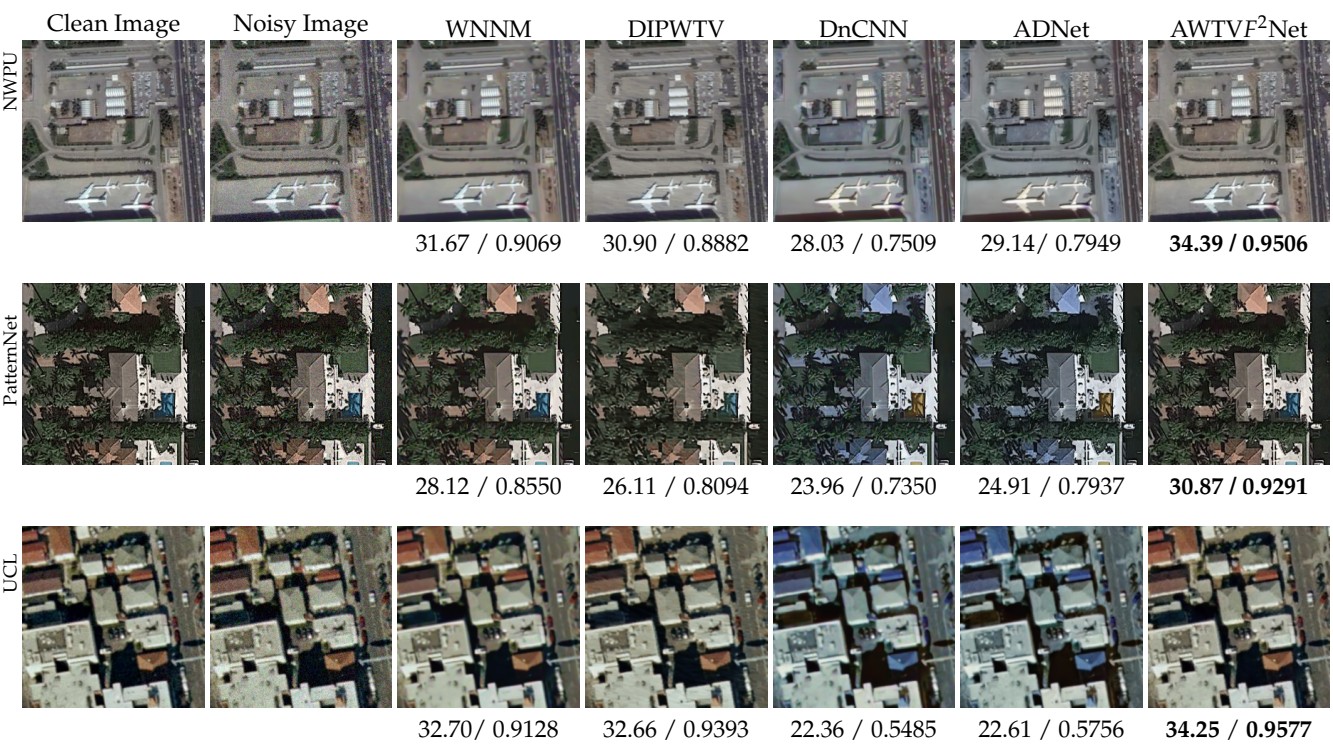

**Figure 8.** Restored images of NWPU, PatternNet and UCL datasets with noise level $\sigma^2 = 15$.

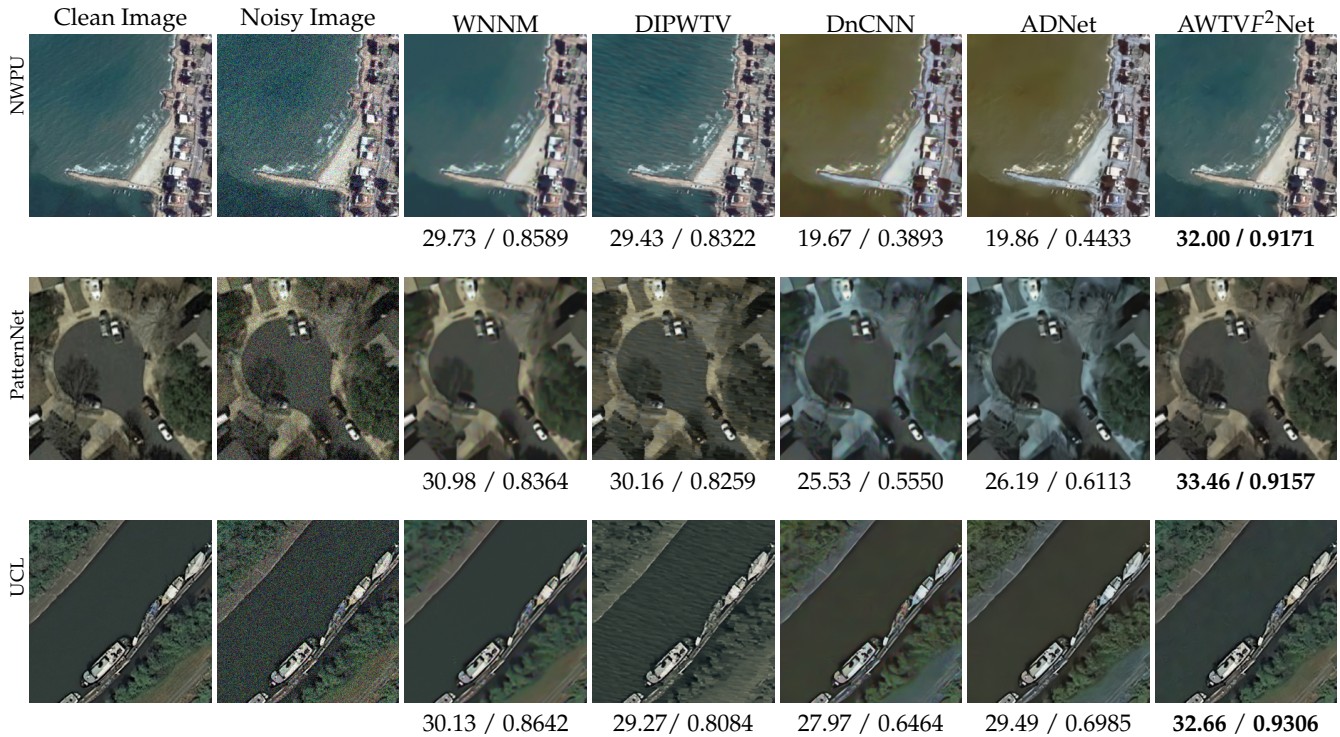

**Figure 9.** Denoised images in NWPU, PatternNet and UCL datasets with noise level $\sigma^2 = 25$.

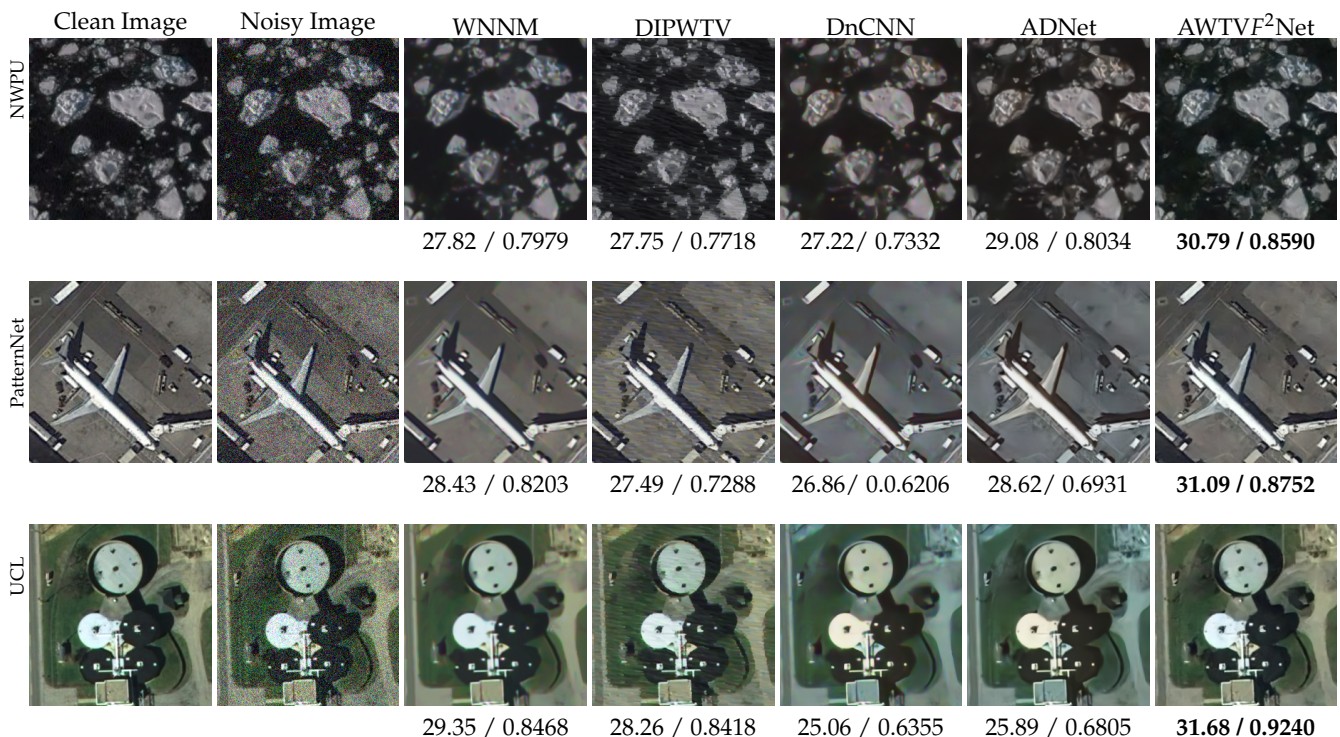

**Figure 10.** Restored results on NWPU, PatternNet and UCL datasets with noise level $\sigma^2 = 35$.

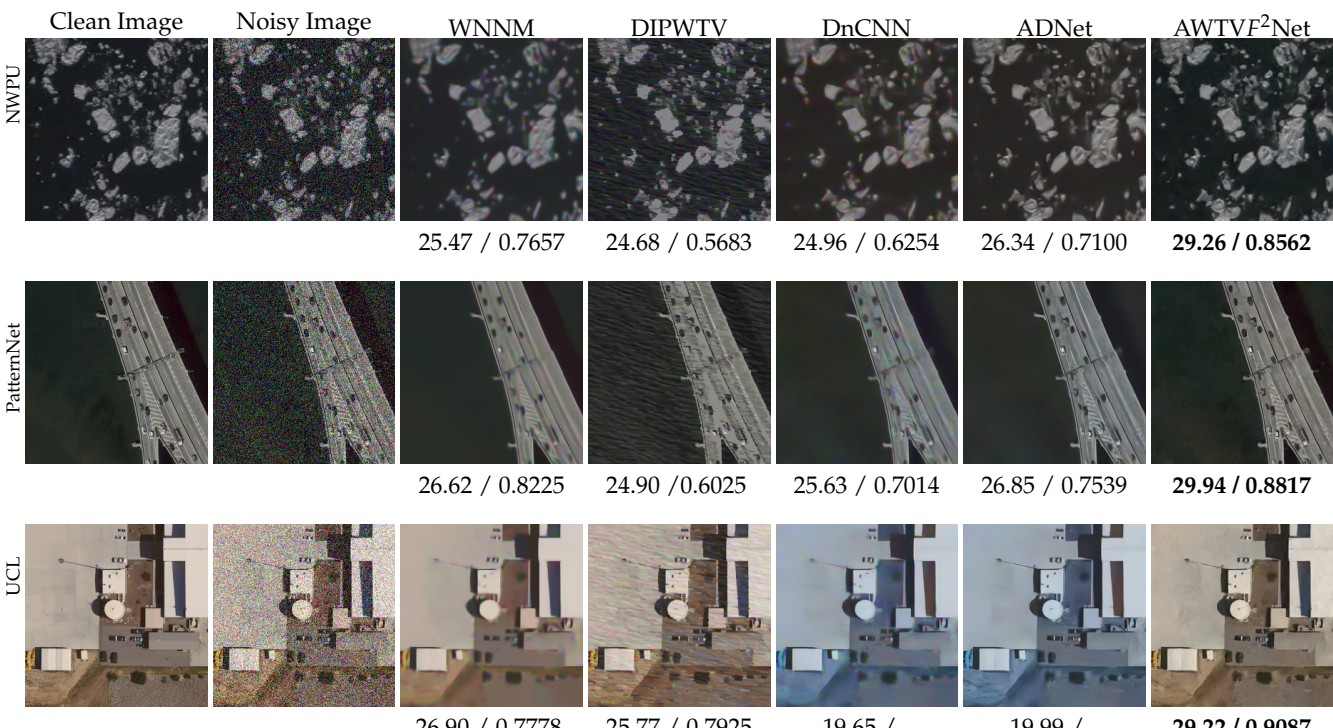

**Figure 11.** Visual results on NWPU, PatternNet and UCL datasets with noise level $\sigma^2 = 50$.

We provide the average PSNR and SSIM values of the different methods on the NWPU, PatternNet, and UCL datasets at the four different noise levels in Table 1. Notably, AWTV$F^2$Net obtained the highest PSNR and SSIM values on NWPU and PatternNet datasets at all noise levels. BM3D had the highest PSNR value on the UCL dataset with a noise level $\sigma^2 = 15$, while our method had the highest SSIM value. AWTV$F^2$Net also obtained the highest PSNR and SSIM values for the other noise levels on the UCL datasets. Overall, whether from the visual comparison or the numerical comparison, our proposed AWTV$F^2$Net presented the best performance and retained more important fine image contents in the denoised images.

To further verify the superior performance of our approach, we randomly selected a class of image from each dataset and enlarged detailed image content to compare the restored image quality of different methods. Figure 12 shows the comparison results of all approaches on the 'Commercial area' image with white Gaussian noise level $\sigma^2 = 50$. Figure 12 indicates that our proposed AWTV$F^2$ had the highest PSNR and SSIM values in this case. From the enlarged region of each image, we can see that DIPTV and DIPWTV obtained restored images which were fuzzy. Meanwhile, some water ripples existed in the image denoised by the DIPWTV approach. The WNNM, DnCNN, and ECDNet methods over-smoothed the edges of buildings. The BM3D approach was better than WNNM, DnCNN, and ECDNet, but it still smoothed the edges of buildings. The ADNet method obtained a competitive restored image with slight color distortion. In contrast, the image denoised by our method not only effectively removed noise but also prevented detailed image contents from smoothing.

We used ten classes of images from the NWPU dataset, and the average PSNR values of different methods with four noise levels for each class of images are provided in Table 2. The table shows that our model had the highest PSNR values on these classified images with a noise levels of $\sigma^2 = 15$, except for the 'Desert' image, for which BM3D obatined the highest PSNR. For noise levels of $\sigma^2 = \{25, 50\}$, AWTV$F^2$Net also obtained the highest PSNR values among all ten classes of images. As for the noise level $\sigma^2 = 35$, the BM3D approach obtained a PSNR value of 27.60—the biggest among all compared methods. Combining the visual and numerical results, our proposed AWTV$F^2$Net outperformed compared approaches.

In the same way, we compared our method with other approaches on the PatternNet dataset. First, we compared enlarged details of the 'Harbor' image with noise level $\sigma^2 = 50$. The enlarged details of different methods are shown in Figure 13. From the figure, it can be seen that AWTV$F^2$Net had the highest PSNR and SSIM values (located under each image). According to the enlarged regions, The WNNM made the restored image more blurred; the BM3D method smoothed some image contents in the denoised image; the DIPTV and DIPWTV models restored images with some water ripples, andADNet, DnCNN, and ECDNet approaches effectively removed noise, but also brought color distortion to these images, which was especially serious for the DnCNN and ECDNet methods. A numerical comparisons of the average PSNR values for each class of images from the PatternNet dataset with noise levels $\sigma^2 = \{15, 25, 35, 50\}$ is shown in Table 3. We selected ten image classes for testing, where each class contained five samples from the PatternNet dataset.

Table 3 shows that our model had the highest PSNR values for all the classes among compared approaches. Notably, the BM3D model obtained competitive PSNR values at the noise levels $\sigma^2 = \{15, 25\}$, presenting a small gap from our model's performance. Combining the visual comparisons from Figure 13 and the numerical results, our model outperformed the compared methods on the PatternNet dataset.

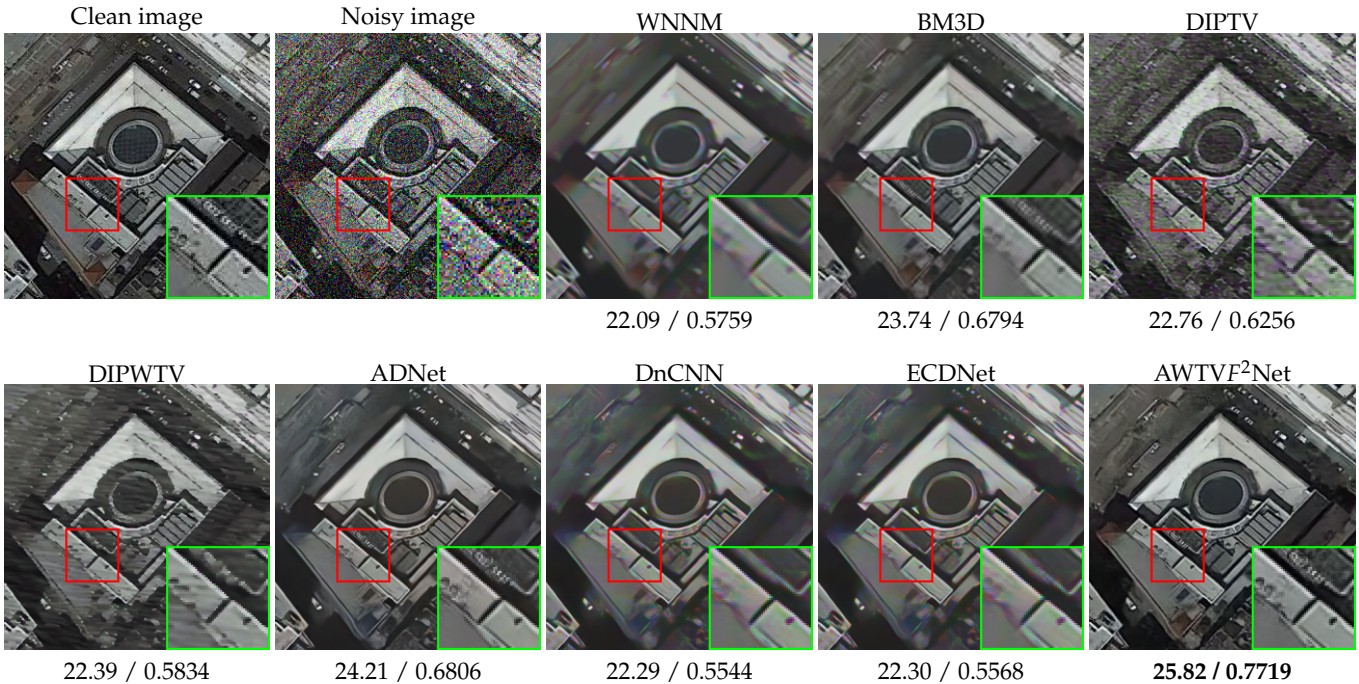

**Figure 12.** Detailed comparisons on the 'Commercial area' with noise level $\sigma^2 = 50$.

**Table 2.** PSNR comparisons of all classes' images in the NWPU dataset. The bold values denote the best results.

| Image | Aitplane | Baseball Diamond | Beach | Church | Commercial Area | Desert | Mountain | Palace | Roundabout | Sea Ice |
|---|---|---|---|---|---|---|---|---|---|---|
| Noise level | | | | | 15 | | | | | |
| WNNM | 31.67 | 31.27 | 29.64 | 30.93 | 28.88 | 31.31 | 29.82 | 31.52 | 28.75 | 31.17 |
| BM3D | 34.09 | 33.68 | 32.42 | 33.43 | 31.77 | **33.80** | 32.41 | 33.80 | 31.68 | 33.69 |
| ADNet | 29.14 | 26.08 | 20.06 | 30.62 | 30.31 | 15.57 | 20.77 | 24.60 | 28.33 | 31.38 |
| DIPWTV | 28.03 | 25.48 | 19.82 | 29.06 | 28.00 | 15.50 | 20.46 | 24.13 | 26.59 | 29.57 |
| DIPTV | 28.03 | 25.48 | 19.82 | 29.05 | 27.97 | 15.50 | 20.46 | 24.13 | 26.58 | 29.57 |
| DnCNN | 29.74 | 30.47 | 28.26 | 28.87 | 25.90 | 28.13 | 29.01 | 30.25 | 27.30 | 30.36 |
| ECDNet | 30.90 | 31.47 | 29.06 | 29.62 | 26.82 | 30.64 | 29.63 | 31.16 | 28.44 | 31.87 |
| AWTV$F^2$Net | **34.39** | **34.29** | **32.84** | **33.85** | **32.28** | 30.43 | **32.91** | **34.00** | **32.25** | **33.74** |
| Noise level | | | | | 25 | | | | | |
| WNNM | 28.35 | 28.77 | 29.73 | 28.28 | 26.12 | 30.74 | 28.08 | 28.77 | 28.54 | 28.54 |
| BM3D | 30.63 | 30.93 | 31.92 | 30.43 | 28.43 | 32.78 | 30.17 | 31.01 | 30.83 | 30.83 |
| ADNet | 28.83 | 25.60 | 19.86 | 29.11 | 28.12 | 19.10 | 26.81 | 24.39 | 28.51 | 28.51 |
| DIPWTV | 27.78 | 28.81 | 29.43 | 28.02 | 25.36 | 29.40 | 27.52 | 28.53 | 28.60 | 28.60 |
| DIPTV | 27.78 | 28.66 | 29.15 | 28.08 | 25.25 | 28.61 | 27.86 | 28.87 | 28.64 | 28.64 |
| DnCNN | 27.07 | 24.77 | 19.66 | 27.38 | 25.87 | 18.98 | 25.55 | 23.61 | 26.88 | 26.88 |
| ECDNet | 27.06 | 24.77 | 19.65 | 27.39 | 25.84 | 18.98 | 25.55 | 23.61 | 26.88 | 26.88 |
| AWTV$F^2$Net | **31.11** | **31.29** | **32.00** | **30.98** | **29.13** | **29.34** | **30.42** | **31.34** | **31.25** | **31.25** |
| Noise level | | | | | 35 | | | | | |
| WNNM | 26.59 | 26.58 | 28.87 | 26.45 | 26.10 | 25.79 | 28.20 | 26.97 | 26.66 | 27.82 |
| BM3D | 28.58 | 28.41 | 30.72 | 28.17 | 27.94 | **27.60** | 30.10 | 28.98 | 28.73 | 29.73 |
| ADNet | 27.70 | 22.48 | 20.36 | 27.67 | 27.13 | 16.51 | 25.01 | 24.14 | 27.38 | 29.08 |
| DIPWTV | 26.29 | 26.50 | 27.90 | 26.31 | 25.27 | 25.10 | 26.43 | 26.47 | 26.64 | 27.75 |
| DIPTV | 26.54 | 26.49 | 27.19 | 26.57 | 25.81 | 25.18 | 25.65 | 26.70 | 26.66 | 28.11 |
| DnCNN | 25.84 | 21.87 | 20.13 | 25.99 | 25.25 | 16.28 | 24.20 | 23.11 | 25.59 | 27.22 |
| ECDNet | - | - | - | - | - | - | - | - | - | - |
| AWTV$F^2$Net | **29.50** | **29.35** | **31.13** | **29.21** | **28.98** | 26.45 | **30.21** | **29.64** | **29.73** | **30.79** |
| Noise level | | | | | 50 | | | | | |
| WNNM | 23.75 | 25.15 | 27.02 | 24.24 | 24.05 | 27.79 | 26.11 | 24.92 | 24.33 | 25.47 |
| BM3D | 25.60 | 27.00 | 28.74 | 25.57 | 25.54 | 28.71 | 28.11 | 27.03 | 26.07 | 26.77 |
| ADNet | 24.63 | 24.26 | 20.37 | 25.54 | 25.34 | 19.05 | 26.00 | 23.67 | 25.53 | 26.34 |
| DIPWTV | 23.39 | 24.27 | 24.77 | 23.99 | 23.08 | 23.79 | 24.51 | 24.22 | 24.33 | 24.68 |
| DIPTV | 23.44 | 22.79 | 23.26 | 23.59 | 23.32 | 22.25 | 24.10 | 23.78 | 23.31 | 23.97 |
| DnCNN | 22.93 | 23.16 | 20.01 | 24.15 | 23.52 | 18.82 | 24.47 | 22.43 | 23.88 | 24.96 |
| ECDNet | 22.92 | 23.17 | 20.01 | 24.13 | 23.54 | 18.84 | 24.49 | 22.42 | 23.86 | 24.97 |
| AWTV$F^2$Net | **27.03** | **28.26** | **29.69** | **28.03** | **27.48** | **28.91** | **29.43** | **28.16** | **28.11** | **29.26** |

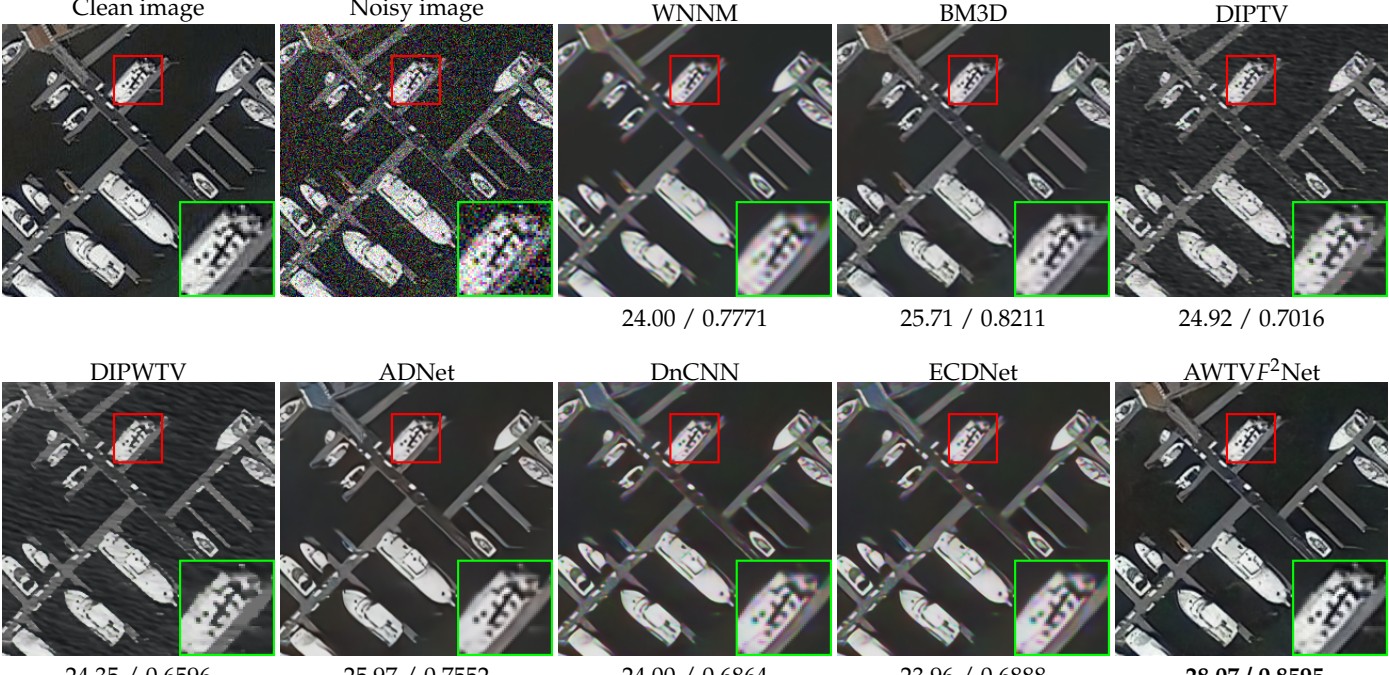

**Figure 13.** Detailed comparisons on the 'Harbor' image with noise level $\sigma = 50$.

**Table 3.** PSNR results of different methods for each calss from the PatternNet dataset. The bold values denote the best results.

| Image | Aitplane | Bridge | Cemetery | Closedroad | Coastalmansion | Footballfield | Harbor | Shippingyard | Swimmingpool | Wastewaterplant |
|---|---|---|---|---|---|---|---|---|---|---|
| Noise level | | | | | 15 | | | | | |
| WNNM | 32.18 | 32.28 | 30.12 | 30.39 | 28.12 | 32.85 | 31.65 | 30.56 | 30.80 | 32.60 |
| BM3D | 33.42 | 32.59 | 31.55 | 32.31 | 30.80 | 34.71 | 33.68 | 33.08 | 32.74 | 34.67 |
| ADNet | 28.09 | 29.02 | 25.64 | 24.80 | 24.91 | 23.77 | 26.39 | 24.67 | 25.88 | 23.93 |
| DIPWTV | 5.96 | 30.69 | 30.16 | 29.38 | 26.11 | 32.31 | 30.84 | 28.74 | 30.72 | 32.45 |
| DIPTV | 15.63 | 29.46 | 28.90 | 28.62 | 25.22 | 31.46 | 29.58 | 27.13 | 29.82 | 31.38 |
| DnCNN | 27.25 | 28.09 | 24.93 | 24.24 | 23.96 | 23.48 | 25.74 | 24.02 | 25.25 | 23.62 |
| ECDNet | 27.25 | 28.07 | 24.93 | 24.23 | 23.95 | 23.48 | 25.74 | 24.01 | 25.26 | 23.62 |
| AWTV$F^2$Net | **34.32** | **33.40** | **31.64** | **32.35** | **30.87** | **34.75** | **33.71** | **33.46** | **32.76** | **34.71** |
| Noise level | | | | | 25 | | | | | |
| WNNM | 30.20 | 31.61 | 27.52 | 31.16 | 25.71 | 31.35 | 28.95 | 28.56 | 29.09 | 31.35 |
| BM3D | 32.31 | 33.17 | 29.70 | 33.51 | 27.96 | 33.23 | 31.21 | 30.49 | 30.72 | 33.46 |
| ADNet | 29.60 | 32.27 | 25.01 | 27.30 | 22.09 | 25.50 | 25.97 | 22.85 | 21.90 | 25.12 |
| DIPWTV | 29.72 | 30.53 | 27.80 | 30.13 | 25.58 | 29.96 | 28.47 | 28.15 | 28.76 | 30.63 |
| DIPTV | 29.69 | 30.27 | 27.85 | 30.23 | 25.40 | 29.64 | 28.54 | 28.19 | 13.17 | 30.43 |
| DnCNN | 28.04 | 30.57 | 24.01 | 26.54 | 21.32 | 24.97 | 25.04 | 22.35 | 21.46 | 24.63 |
| ECDNet | 28.03 | 30.59 | 24.00 | 26.55 | 21.31 | 24.97 | 25.04 | 22.36 | 21.46 | 24.64 |
| AWTV$F^2$Net | **32.64** | **33.23** | **29.93** | **33.79** | **28.48** | **33.39** | **31.65** | **30.76** | **31.11** | **33.73** |
| Noise level | | | | | 35 | | | | | |
| WNNM | 28.43 | 29.60 | 25.78 | 28.98 | 23.86 | 29.53 | 27.07 | 26.73 | 26.10 | 29.36 |
| BM3D | 30.43 | 30.77 | 27.54 | 31.42 | 25.81 | 31.46 | 29.07 | 28.68 | 28.08 | 31.23 |
| ADNet | 28.62 | 30.35 | 24.34 | 26.81 | 21.59 | 25.32 | 25.61 | 22.63 | 21.59 | 24.75 |
| DIPWTV | 27.49 | 28.28 | 25.92 | 27.49 | 24.13 | 26.74 | 26.89 | 25.41 | 4.93 | 27.92 |
| DIPTV | 27.01 | 27.95 | 25.67 | 26.89 | 24.64 | 26.62 | 27.04 | 26.70 | 26.72 | 26.86 |
| DnCNN | 26.86 | 28.77 | 23.17 | 25.82 | 20.65 | 24.59 | 24.43 | 21.97 | 20.93 | 24.10 |
| ECDNet | - | - | - | - | - | - | - | - | - | - |
| AWTV$F^2$Net | **31.09** | **31.23** | **28.56** | **32.23** | **27.03** | **31.75** | **30.01** | **29.37** | **29.16** | **32.18** |
| Noise level | | | | | 50 | | | | | |
| WNNM | 26.12 | 26.73 | 23.86 | 26.60 | 21.77 | 27.41 | 24.77 | 24.30 | 24.48 | 26.96 |
| BM3D | 28.13 | 27.13 | 25.31 | 28.90 | 23.36 | 29.24 | 26.39 | 26.26 | 26.54 | 28.47 |
| ADNet | 27.29 | 27.06 | 23.47 | 25.72 | 20.82 | 24.91 | 24.64 | 22.13 | 21.26 | 23.99 |
| DIPWTV | 24.47 | 25.15 | 23.42 | 24.87 | 22.38 | 24.77 | 24.50 | 24.13 | 6.38 | 25.12 |
| DIPTV | 23.05 | 23.70 | 22.52 | 23.44 | 22.71 | 23.34 | 24.21 | 23.75 | 24.10 | 23.04 |
| DnCNN | 25.43 | 26.18 | 22.25 | 24.63 | 19.79 | 23.95 | 23.23 | 21.22 | 20.38 | 23.33 |
| ECDNet | 25.41 | 26.09 | 22.25 | 24.65 | 19.78 | 23.95 | 23.22 | 21.23 | 20.39 | 23.31 |
| AWTV$F^2$Net | **29.63** | **30.81** | **27.06** | **30.60** | **25.52** | **30.31** | **28.54** | **28.00** | **28.27** | **30.64** |

Figure 14 shows visual comparisons of different models on the 'Dense residential' image, which is from the UCL dataset with noise level $\sigma^2 = 50$. The PSNR and SSIM (located under each image) indicate that the values for our method were highest among the compared models. From each enlarged region, we can see that WNNM and BM3D removed the noise without preserving the small contents in images. The DIPTV and DIPWTV models could not remove the noise, and also caused some water ripples in denoised images. The ADNet, DnCNN, and ECDNet models not only over-smoothed the restored images, but also brought some color distortion, leading to the color of buildings being changed. Our AWTV$F^2$Net obtained a higher quality restored image while preserving detailed contents, compared to the other approaches.

We selected eight classes of images from the UCL dataset, where each class had five samples. Using these images with four different noise levels $\sigma^2 = \{15, 25, 35, 50\}$, we tested the performance of the compared models. The average PSNR values for each class are presented in Table 4. We can see that AWTV$F^2$Net obtained the highest PSNR value for the noise levels of 15 and 50; while, for the noise level of 25, the BM3D method had the highest value (0.18 higher than that of our model). However, AWTV$F^2$Net had the highest PSNR values for other classes. As for the noise level of 35, the BM3D model outperformed our proposed method on the 'Freeway' image, with the PSNR value of the BM3D approach on the 'Freeway' image being 0.15 higher than that of AWTV$F^2$Net. Our model obtained the best performance for the other classes of images.

Overall, in both visual and numerical comparisons, our proposed AWTV$F^2$Net showed superior performance over the other compared methods in the additive white Gaussian noise experiments. Our model thus provides effective noise removal along with powerful preservation of fine image details.

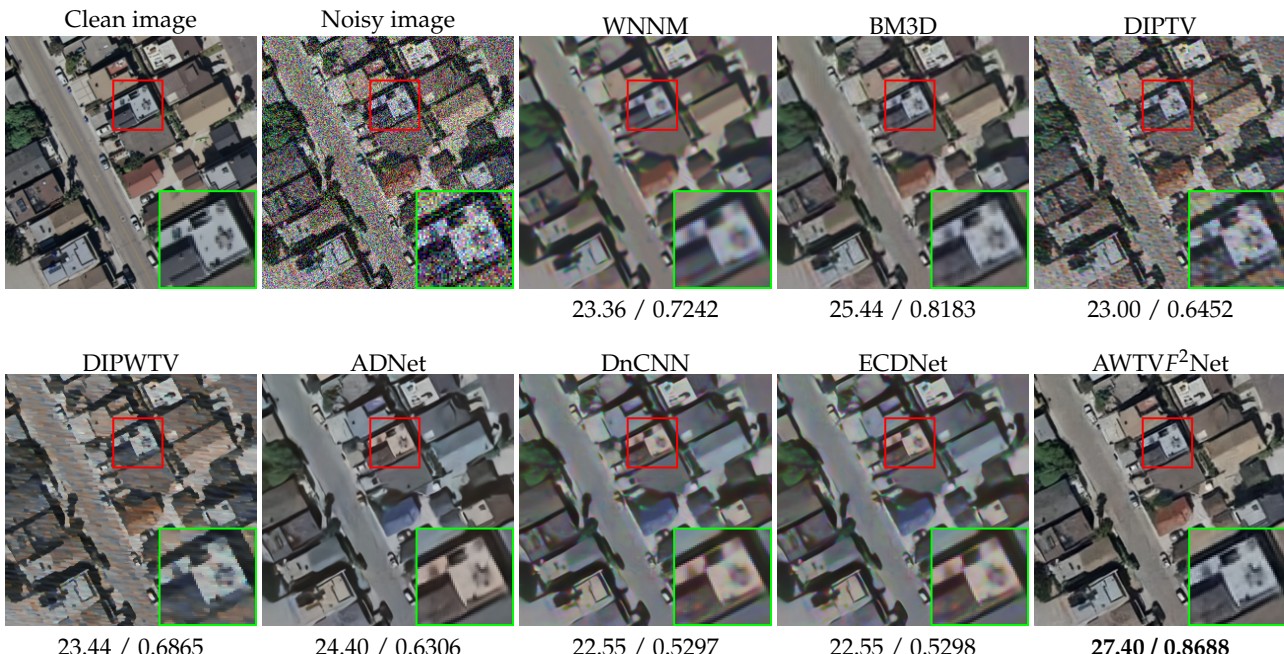

**Figure 14.** Detailed comparisons of the 'Dense residential' image with $\sigma = 50$.

**Table 4.** Comparison results of different methods on UCL dataset. The bold values denote the best results.

| Image | Aitplane | Buildings | Denseresidential | Forest | Freeway | Intersection | Rvier | Storagetanks |
|---|---|---|---|---|---|---|---|---|
| Noise level | | | | 15 | | | | |
| WNNM | 32.21 | 32.70 | 31.17 | 26.94 | 31.95 | 32.22 | 32.98 | 34.16 |
| BM3D | 32.74 | 34.17 | 32.03 | 27.81 | 32.07 | 33.31 | 34.44 | 35.22 |
| ADNet | 19.11 | 22.61 | 19.34 | 25.20 | 30.28 | 28.23 | 30.38 | 26.50 |
| DIPWTV | 31.81 | 32.66 | 6.05 | 24.64 | 31.05 | 31.59 | 32.47 | 33.62 |
| DIPTV | 31.48 | 31.62 | 16.59 | 24.63 | 30.20 | 30.89 | 30.14 | 32.72 |
| DnCNN | 19.02 | 22.36 | 19.18 | 23.86 | 29.11 | 27.43 | 29.29 | 26.12 |
| ECDNet | 19.01 | 22.36 | 19.18 | 23.86 | 29.09 | 27.42 | 29.27 | 26.12 |
| AWTV$F^2$Net | **33.06** | **34.25** | **32.09** | **28.00** | **32.10** | **33.45** | **34.84** | **35.29** |
| Noise level | | | | 25 | | | | |
| WNNM | 27.80 | 29.67 | 27.97 | 23.99 | 27.64 | 29.23 | 30.13 | 26.92 |
| BM3D | 29.63 | 31.33 | 29.71 | **26.56** | 28.38 | 31.03 | 32.50 | 28.39 |
| ADNet | 22.69 | 22.46 | 26.27 | 23.95 | 27.73 | 27.50 | 29.49 | 27.20 |
| DIPWTV | 27.39 | 29.40 | 28.00 | 23.03 | 26.39 | 29.20 | 29.27 | 20.98 |
| DIPTV | 27.23 | 29.49 | 27.81 | 23.86 | 26.57 | 29.31 | 29.42 | 25.43 |
| DnCNN | 22.13 | 22.01 | 24.98 | 22.29 | 26.16 | 26.30 | 27.97 | 25.43 |
| ECDNet | 22.12 | 22.01 | 24.96 | 22.28 | 26.12 | 26.30 | 27.96 | 25.37 |
| AWTV$F^2$Net | **29.90** | **31.74** | **29.79** | 26.38 | **28.67** | **31.05** | **32.66** | **28.45** |
| Noise level | | | | 35 | | | | |
| WNNM | 25.86 | 27.41 | 26.73 | 25.38 | 28.42 | 27.42 | 28.11 | 29.35 |
| BM3D | 27.49 | 29.26 | 28.53 | 26.77 | **29.58** | 29.47 | 30.31 | 31.10 |
| ADNet | 22.43 | 22.25 | 19.23 | 25.51 | 28.35 | 26.99 | 28.51 | 25.89 |
| DIPWTV | 25.85 | 26.62 | 6.05 | 24.67 | 5.65 | 27.27 | 27.59 | 28.26 |
| DIPTV | 25.98 | 26.74 | 16.59 | 24.25 | 27.01 | 27.30 | 27.97 | 27.37 |
| DnCNN | 21.65 | 21.59 | 18.83 | 24.26 | 26.84 | 25.47 | 26.76 | 25.06 |
| ECDNet | - | - | - | - | - | - | - | - |
| AWTV$F^2$Net | **28.36** | **30.42** | **28.57** | **26.96** | 29.43 | **29.87** | **31.11** | **31.68** |
| Noise level | | | | 50 | | | | |
| WNNM | 23.73 | 24.76 | 24.59 | 24.27 | 26.90 | 25.27 | 25.97 | 26.73 |
| BM3D | 25.20 | 26.80 | 26.56 | 25.66 | 28.22 | 27.37 | 27.77 | 28.59 |
| ADNet | 21.90 | 21.87 | 19.14 | 24.57 | 27.42 | 26.05 | 26.92 | 25.21 |
| DIPWTV | 23.75 | 24.09 | 6.05 | 23.07 | 25.11 | 25.02 | 25.22 | 25.08 |
| DIPTV | 22.89 | 23.18 | 16.59 | 21.76 | 23.66 | 23.97 | 23.79 | 23.19 |
| DnCNN | 20.98 | 20.93 | 18.56 | 23.45 | 25.79 | 24.24 | 25.29 | 24.16 |
| ECDNet | 20.96 | 20.93 | 18.56 | 23.48 | 25.75 | 24.26 | 25.25 | 24.17 |
| AWTV$F^2$Net | **27.20** | **28.75** | **27.15** | **25.90** | **28.60** | **28.48** | **29.63** | **30.29** |

### 5.2. Experimental Results of Mixture Noise

To further test the performance of our model, we conducted comprehensive experiments using multiple kinds of mixed noise, such as Speckle noise, white Gaussian noise, and Salt&Pepper noise. We use the same (NWPU, PatternNet, and UCL) datasets to compare the performances of all methods. The noisy images include three kinds of mixed noise. The first type of noise was a mixture of white Gaussian and Salt&Pepper noise, the second type was a mixture of white Gaussian and Speckle noise, and the final type was a mixture of Salt&Pepper and Speckle noise. Figures 15–17 show the restored images obtained by DIPTV, DIPWTV, DnCNN-B, and our proposed method for the three types of mixed noise, respectively, in the NWPU, PatternNet, and UCL datasets.

Figure 15 shows denoised results by different methods. The images used for Figure 15 were distorted by the mixture of white Gaussian noise and Salt&Pepper. This type of noise was introduced in Section 4.1. According to Figure 15, it can be seen that our model had the highest quality of restored images, as it presented the highest PSNR and SSIM values. Meanwhile, the DnCNN-B approach could not remove Salt&Pepper noise, and brought some color distortion into the denoised images. The DIPWTV and DIPTV methods also could not obtain good visual quality of the restored images. In contrast, our model obtained higher quality denoised images while preserving fine features.

Figure 16 shows the denoised images of compared approaches for the mixed white Gaussian and Speckle noise. It is evident that all methods could not obtain good visually restored images. However, our model outperformed the others in this situation, having higher PSNR and SSIM values.

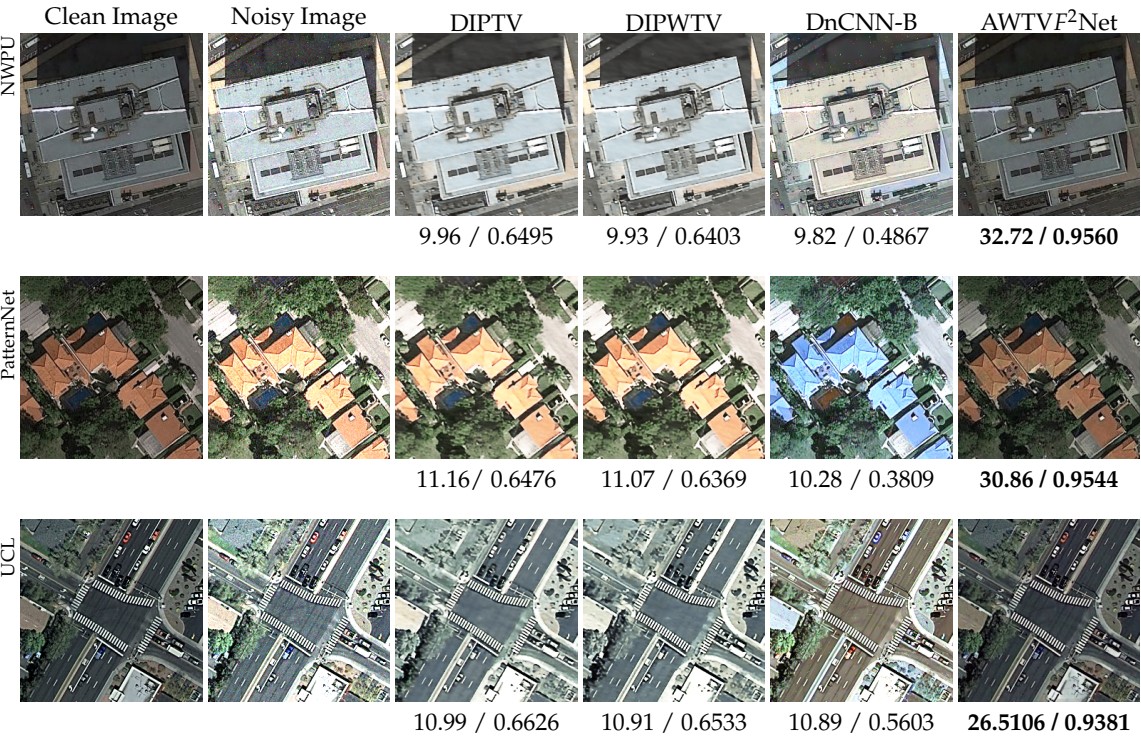

**Figure 15.** Restored images distorted by mixture noise of Gaussian and Salt&Pepper.

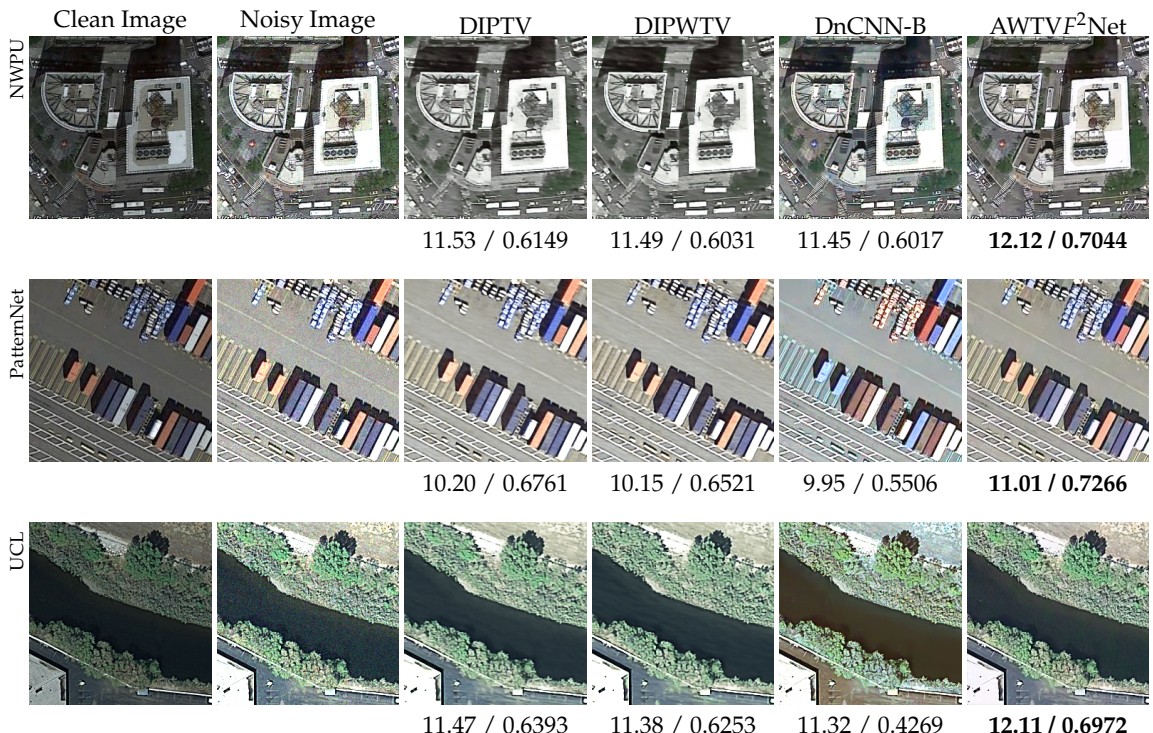

**Figure 16.** Denoised images distorted by mixture noise of Gaussian and Speckle.

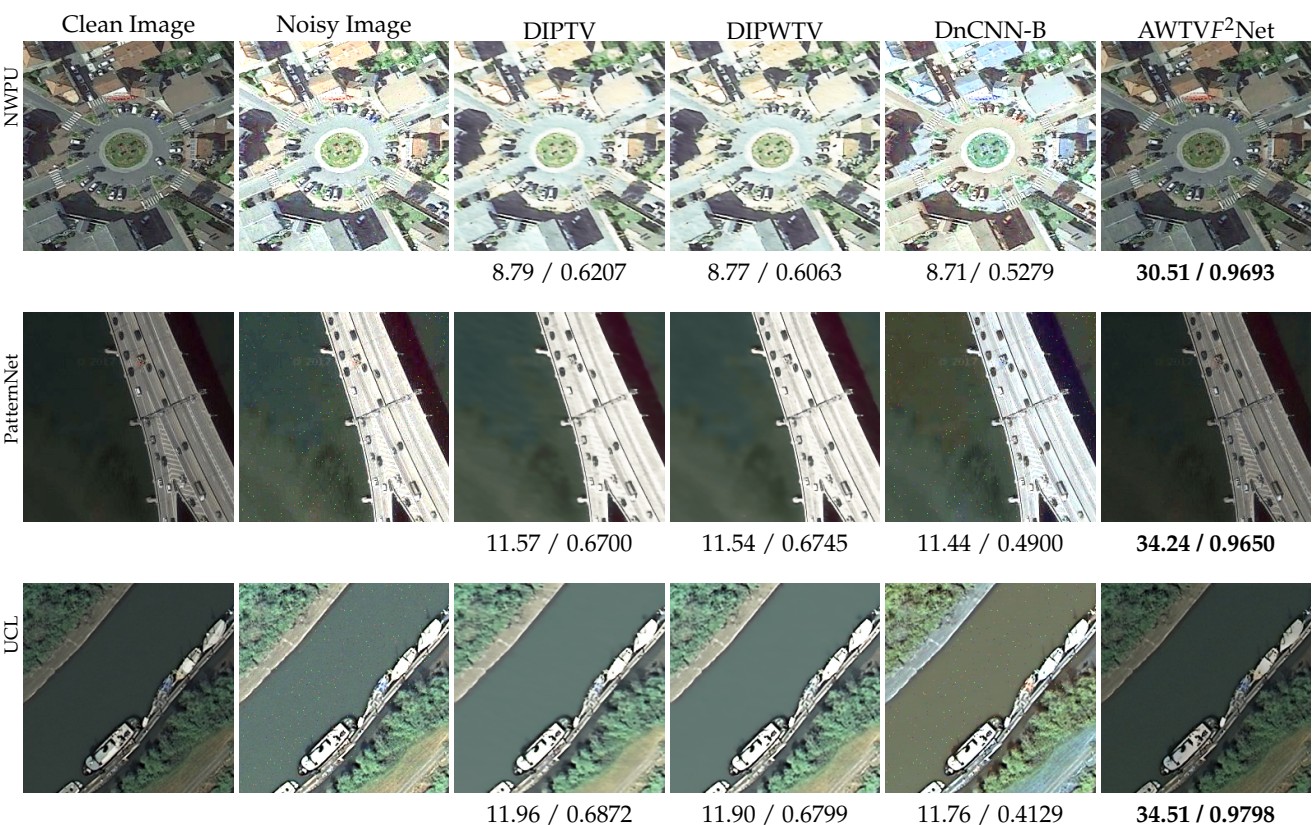

**Figure 17.** Restored results of images distorted by mixture noise of Salt&Pepper and Speckle.

Although the DnCNN model had a competitive capability to remove noise, it caused color distortion and smoothed the details of images, as shown in Figure 16. For this case of noise, the DIPTV and DIPWTV models caused some water ripples in the restored images.

Figure 17 presents the experimental results with the mixed Salt&Pepper and Speckle noise. According to the visual comparisons in Figure 17, we can see that the DIPTV and DIPWTV approaches could not remove noise artifacts in the second row of Figure 17. Meanwhile, they over-smoothed the denoised images, causing image details to be smoothed. The DnCNN-B model still could not handle Salt&Pepper noise. However, our model makes up for these issues of the compared methods. The proposed AWTV$F^2$Net not only smoothed the noise and noise artifacts, but also effectively preserved the detail contents of images. We present the overall results for the three types of mixed noise in Table 5. The left values denote the PSNR, and the right values are the SSIM. These results prove that our model obtained the best performance.

**Table 5.** Numercial comparisons of different datasets with various types of mixture noise. The bold values denote the best results.

| Datasets | Method | DIPTV | DIPWTV | DnCNN-B | AWTV$F^2$Net |
|---|---|---|---|---|---|
| Metrics [1] | Noise Type | P/S | P/S | P/S | P/S |
| NWPU | Gaussian + Salt&Pepper | 9.65 / 0.5716 | 9.50 / 0.5458 | 9.64 / 0.3765 | **29.04 / 0.9127** |
| | Gaussian + Speckle | 9.65 / 0.5717 | 9.45 / 0.5355 | 9.68 / 0.3899 | **11.71 / 0.6709** |
| | Salt&Pepper + Speckle | 9.63 / 0.5744 | 9.49 / 0.5497 | 9.64 / 0.3966 | **28.90 / 0.9193** |
| PatternNet | Gaussian+Salt&Pepper | 10.76 / 0.6389 | 10.71 / 0.6183 | 10.47 / 0.4065 | **29.98 / 0.9297** |
| | Gaussian + Speckle | 10.78 / 0.6402 | 10.54 / 0.5961 | 10.52 / 0.4251 | **11.24 / 0.6639** |
| | Salt&Pepper + Speckle | 10.76 / 0.6428 | 10.51 / 0.5902 | 10.47 / 0.4249 | **29.59 / 0.9311** |
| UCL | Gaussian+Salt&Pepper | 9.10 / 0.5343 | 8.82 / 0.4729 | 9.31 / 0.3944 | **26.08 / 0.8520** |
| | Gaussian + Speckle | 9.11 / 0.5351 | 8.67 / 0.4620 | 9.33 / 0.4035 | **13.30 / 0.6608** |
| | Salt&Pepper + Speckle | 9.09 / 0.5343 | 8.57 / 0.4490 | 9.31 / 0.4128 | **25.26 / 0.8425** |

We further compared the performance of our model with other methods by enlarging details of images. Figures 18–20 present the restored results for different classes from the three datasets with the mixture of Salt&Pepper and Speckle noise. Figure 18 shows enlarged detailed comparisons to other methods for the 'Commercial area' image from the NWPU dataset. Notably, the DIPTV and DIPWTV models presented over-smoothed denoised images. It is obvious that the edges of the target in the enlarged regions were smoothed. While the DnCNN-B model kept the edges of the target, it could not remove the Salt&Pepper noise. It can be seen, from the restored image of our proposed method, that it effectively removed the mixed noise without losing the target's edge information. We also present the overall PSNR values for 10 classes of images with all three types of mixed noise in Table 6, for images included in the NWPU dataset. From Table 6, we can conclude that our proposed AWTV$F^2$Net outperformed other methods, as it obtained the highest PSNR values.

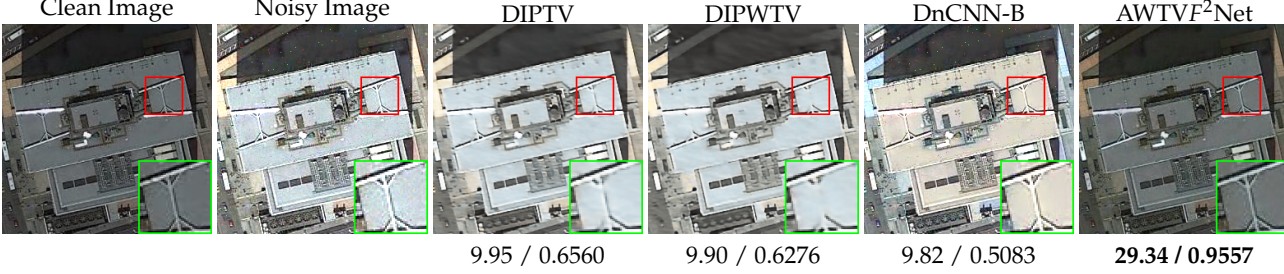

**Figure 18.** Enlarged details of the 'Commercial area' image with the third case of mixture noise.

**Table 6.** PSNR comparisons of each class in the NWPU dataset with various types of mixture noise. The bold values denote the best results.

| Image | Aitplane | Baseball Diamond | Beach | Church | Commercial Area | Desert | Mountain | Palace | Roundabout | Sea Ice |
|---|---|---|---|---|---|---|---|---|---|---|
| Noise type | | | | | Gaussian + Salt&Pepper | | | | | |
| DIPTV | 8.52 | 9.07 | 9.42 | 9.85 | 9.96 | 8.66 | 10.80 | 9.74 | 11.23 | 11.99 |
| DIPWTV | 8.47 | 9.03 | 9.40 | 9.81 | 9.93 | 8.65 | 10.78 | 9.69 | 11.21 | 11.90 |
| DnCNN-B | 8.42 | 8.95 | 9.19 | 9.58 | 9.82 | 8.00 | 10.59 | 9.49 | 11.08 | 11.76 |
| AWTV$F^2$Net | **28.05** | **29.75** | **33.26** | **31.78** | **32.72** | **30.62** | **36.58** | **29.67** | **33.78** | **35.69** |
| Noise type | | | | | Gaussian + Speckle | | | | | |
| DIPTV | 6.31 | 9.50 | 10.35 | 11.93 | 11.53 | 7.82 | 7.82 | 11.28 | 8.90 | 15.47 |
| DIPWTV | 4.83 | 6.02 | 10.35 | 11.89 | 11.49 | 4.22 | 7.80 | 11.23 | 8.88 | 15.34 |
| DnCNN-B | 8.48 | 9.29 | 9.23 | 11.81 | 11.45 | 8.20 | 7.75 | 10.68 | 8.81 | 15.14 |
| AWTV$F^2$Net | **17.18** | **10.55** | **12.15** | **12.05** | **12.12** | **21.59** | **13.17** | **11.74** | **12.20** | **15.40** |
| Noise type | | | | | Salt&Pepper + Speckle | | | | | |
| DIPTV | 8.49 | 10.27 | 9.42 | 9.85 | 9.69 | 8.65 | 10.24 | 9.75 | 11.24 | 12.86 |
| DIPWTV | 8.47 | 10.20 | 9.40 | 9.80 | 9.66 | 8.64 | 10.20 | 9.68 | 11.21 | 12.80 |
| DnCNN-B | 8.42 | 10.15 | 9.21 | 9.59 | 9.58 | 8.00 | 9.93 | 9.49 | 11.08 | 12.62 |
| AWTV$F^2$Net | **27.57** | **29.84** | **34.15** | **32.36** | **29.48** | **27.99** | **35.50** | **29.33** | **34.53** | **35.80** |

Figure 19 shows the restored results of the 'Harbor' image from the PatternNet dataset. The visual comparisons in Figure 19 lead to the same conclusions as Figure 18. The DIPTV and DIPWTV models obtained denoised images without preserving details. The DnCNN-B approach led to color distortion in the denoised image without removing the Salt&Pepper noise. Our proposed model removed the mixed noise and prevented detailed image contents from smoothing. We tested all ten classes of images from the PatternNet dataset and the overall PSNR values of images from these classes are provided in Table 7.

From Table 7, we can see that our model obtained the highest PSNR values than compared methods for the first and third cases of mixed noise. For the second case of mixed noise, The DIPTV mode obtained the highest PSNR values in the 'Bridge,' 'Closedroad,' and 'Wastewaterplant' images. In contrast, our model obtained the highest PSNR values in the other classes. Overall, our proposed AWTV$F^2$Net approach obtained a completive performance than the other methods on the PatternNet dataset. In addition, our model preserved more image detail contents than all of the compared approaches.

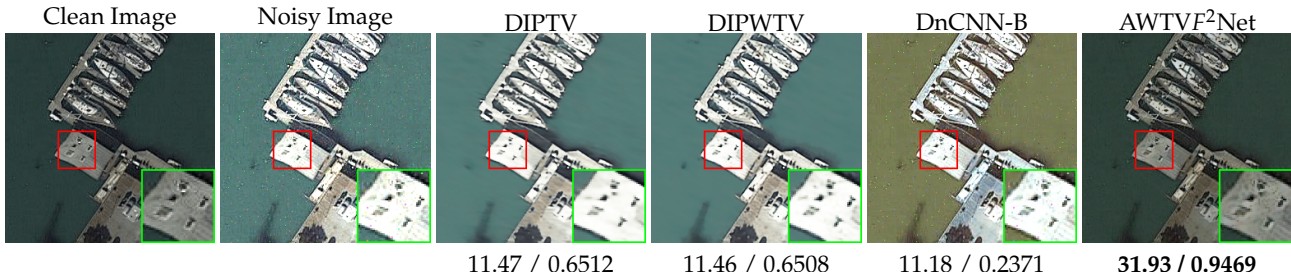

| Clean Image | Noisy Image | DIPTV | DIPWTV | DnCNN-B | AWTV$F^2$Net |
|---|---|---|---|---|---|
| | | 11.47 / 0.6512 | 11.46 / 0.6508 | 11.18 / 0.2371 | **31.93 / 0.9469** |

**Figure 19.** Detailed comparisons of the 'Harbor' image with the third type of mixture noise .

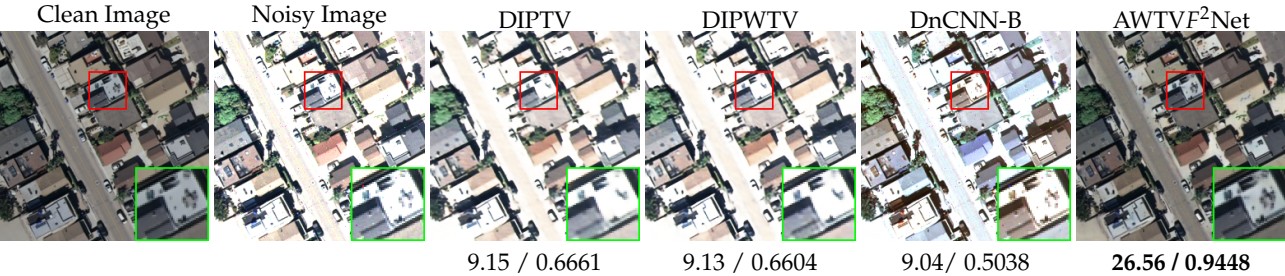

| Clean Image | Noisy Image | DIPTV | DIPWTV | DnCNN-B | AWTV$F^2$Net |
|---|---|---|---|---|---|
| | | 9.15 / 0.6661 | 9.13 / 0.6604 | 9.04 / 0.5038 | **26.56 / 0.9448** |

**Figure 20.** Enlarged details of the 'Dense residential' image with the third type of mixture noise.

**Table 7.** PSNR results of different methods on the PatternNet dataset. The bold values denote the best results.

| Image | Aitplane | Bridge | Cemetery | Closedroad | Coastalmansion | Footballfield | Harbor | Shippingyard | Swimmingpool | Wastewaterplant |
|---|---|---|---|---|---|---|---|---|---|---|
| Noise type | | | | | | Gaussian + Salt&Pepper | | | | |
| DIPTV | 8.51 | 11.57 | 15.16 | 10.81 | 11.16 | 10.96 | 12.98 | 10.90 | 10.74 | 11.05 |
| DIPWTV | 8.50 | 11.55 | 15.09 | 10.79 | 11.07 | 10.95 | 12.95 | 10.85 | 10.70 | 11.02 |
| DnCNN-B | 8.39 | 11.45 | 14.89 | 10.58 | 10.28 | 10.64 | 12.81 | 10.39 | 10.48 | 10.75 |
| AWTV$F^2$Net | **29.51** | **33.86** | **28.56** | **33.73** | **30.86** | **34.20** | **28.56** | **33.25** | **31.66** | **35.09** |
| Noise type | | | | | | Gaussian + Speckle | | | | |
| DIPTV | 8.30 | **13.22** | 11.81 | **10.42** | 12.02 | 9.88 | 11.72 | 10.20 | 10.23 | **10.63** |
| DIPWTV | 5.33 | 13.19 | 15.15 | 10.40 | 8.91 | 9.86 | 11.69 | 10.15 | 10.20 | 10.56 |
| DnCNN-B | 8.27 | 13.12 | 15.09 | 10.21 | 11.75 | 9.45 | 11.46 | 9.95 | 9.73 | 10.24 |
| AWTV$F^2$Net | **12.86** | 13.19 | **12.01** | 10.33 | **12.86** | **10.32** | **11.90** | **11.01** | **11.96** | 10.54 |
| Noise type | | | | | | Salt&Pepper + Speckle | | | | |
| DIPTV | 9.10 | 11.35 | 11.79 | 10.46 | 11.15 | 10.95 | 11.47 | 10.94 | 10.73 | 11.04 |
| DIPWTV | 9.10 | 11.33 | 11.73 | 10.42 | 11.10 | 10.95 | 11.46 | 10.82 | 10.70 | 11.03 |
| DnCNN-B | 9.02 | 11.21 | 11.46 | 10.25 | 10.29 | 10.64 | 11.18 | 10.41 | 10.48 | 10.74 |
| AWTV$F^2$Net | **30.08** | **35.89** | **35.77** | **37.81** | **31.84** | **34.55** | **31.93** | **33.97** | **32.10** | **36.67** |

Figure 20 shows comparisons of enlarged details from the 'Dense residential' image in the UCL dataset for different methods. According to the enlarged regions of each image, the proposed AWTV$F^2$Net model led to a higher quality restored image. The DnCNN-B, DIPTV, and DIPWTV model removed the mixed noise but lost fine feature information of the image. Moreover, the DnCNN-B model caused serious color distortion. For this type of mixed noise, the enlarged detail of the denoised image restored by our model illustrates its superior performance in noise removal and image detail preservation. The numerical results for images from all eight classes in the UCL dataset are shown in Table 8. These results demonstrate that our model outperformed the other approaches in removing all three types of mixed noise.

**Table 8.** Comparison results of different methods on NWPU dataset. The bold values denote the best results.

| Image | Aitplane | Buildings | Denseresidential | Forest | Freeway | Intersection | Rvier | Storagetanks |
|---|---|---|---|---|---|---|---|---|
| Noise type | | | Gaussian + Salt&Pepper | | | | | |
| DIPTV | 8.24 | 9.77 | 9.18 | 8.66 | 5.63 | 8.24 | 11.95 | 9.94 |
| DIPWTV | 8.23 | 9.74 | 9.13 | 8.63 | 6.17 | 8.22 | 11.91 | 9.96 |
| DnCNN-B | 8.12 | 9.54 | 9.05 | 8.55 | 7.26 | 8.08 | 11.76 | 9.82 |
| AWTV$F^2$Net | **27.97** | **27.09** | **27.85** | **31.35** | **31.40** | **29.36** | **32.86** | **27.34** |
| Noise type | | | Gaussian + Speckle | | | | | |
| DIPTV | 7.75 | 11.43 | 6.41 | 8.34 | 6.65 | 8.58 | 11.47 | 9.11 |
| DIPWTV | 4.07 | 11.34 | 4.34 | 8.28 | 7.90 | 5.72 | 11.38 | 9.12 |
| DnCNN-B | 8.73 | 11.69 | 9.38 | 8.27 | 7.91 | 8.43 | 11.32 | 9.01 |
| AWTV$F^2$Net | **21.47** | **17.42** | **14.63** | **14.04** | **19.54** | **13.35** | **12.11** | **14.73** |
| Noise type | | | Salt&Pepper + Speckle | | | | | |
| DIPTV | 8.21 | 9.76 | 9.15 | 8.64 | 5.63 | 8.21 | 11.96 | 9.94 |
| DIPWTV | 8.19 | 9.76 | 9.13 | 8.62 | 6.17 | 8.20 | 11.90 | 9.94 |
| DnCNN-B | 8.10 | 9.55 | 9.04 | 8.54 | 7.22 | 8.07 | 11.76 | 9.81 |
| AWTV$F^2$Net | **27.55** | **25.85** | **26.56** | **31.07** | **29.57** | **28.64** | **34.51** | **28.55** |

Although all methods considered for comparison performed poorly in removing the second type of mixed noise, producing corresponding low PSNR values, our approach outperformed the compared methods in terms of denoising and preserving details. Our proposed model had a superior performance compared to other models in the task of removing the mixed noise. Whether white Gaussian noise or mixed noise is considered, our method can remove noise well without losing detailed information and retain the details of images better than all of the models used for comparison.

### 5.3. Ablation Study

To validate the effectiveness of proposed AWTV$F^2$Net model, our ablation experiments included two aspects: The selections of $\gamma_1$ and $\gamma_2$, and the effect of the SOSB block module in our network.

For the selection of $\gamma_1$ and $\gamma_2$, we fixed the other parameters in our model and set $\gamma_1$ from 0.1 to 0.9, with the constraint of $\gamma_1 + \gamma_2 = 1$. For different settings of $\gamma_1$ and $\gamma_2$, we re-trained our model using the same epoch number. The numerical results under different $\gamma_1$ on the PatternNet dataset are presented in Table 9. We can see that the obtained mean values of PSNR and SSIM were the highest when taking $\gamma_1$ and $\gamma_2$ as 0.4 and 0.6, respectively. In the simulation experiments for our proposed network, we thus assigned the values of $\gamma_1$ and $\gamma_2$ as 0.4 and 0.6. This set of weights can well join the two loss function of $\mathcal{L}_1$ and $\mathcal{L}_2$, allowing our model to obtain superior performance over the compared methods.

**Table 9.** Mean metrics comparisons of different $\gamma_1$, $\gamma_2$ on the PatternNet dataset. The bold values denote the best results.

| $\gamma_1$ | 0.1 | 0.2 | 0.3 | 0.4 | 0.5 | 0.6 | 0.7 | 0.8 | 0.9 |
|---|---|---|---|---|---|---|---|---|---|
| PSNR | 26.78 | 26.74 | 26.73 | **26.88** | 26.56 | 26.71 | 26.60 | 26.49 | 26.24 |
| SSIM | 0.7963 | 0.7893 | 0.7909 | **0.7998** | 0.7875 | 0.7865 | 0.7890 | 0.7953 | 0.7882 |

As for the effect of the SOSB block module, we conducted ablation experiments with two types of networks. One network was our proposed AWTV$F^2$Net, which includes SOSB block modules. The other network did not include the SOSB block modules. We used the same hyperparameters to train the two networks, and tested them on the PatternNet dataset. The mean values of PSNR and SSIM for the network without SOSB block modules were 26.71 and 0.7873, respectively, while our proposed AWTV$F^2$Net with SOSB block modules obtained higher PSNR and SSIM values: 26.88 and 0.7998, respectively. To further illustrate the effect of the SOSB block modules, we provide a visual comparisons of the restored images in Figure 21. These denoised images were obtained by networks with/without SOSB block modules. According to Figure 21, the image restored using the approach without SOSB block modules was denoised, but the image details were over-smoothed (we recommend viewing the small details of the tree in enlarged regions of each image). In contrast, our proposed AWTV$F^2$Net with SOSB block modules not only outperformed better in terms of noise removal, but also preserved some small key details of the image. This is one of the merits of our model over the other compared methods. The values of PSNR and SSIM for Figure 21 are presented in Table 10. Notably, the proposed approach, when deployed with SOSB block modules, obtained higher PSNR and SSIM values.

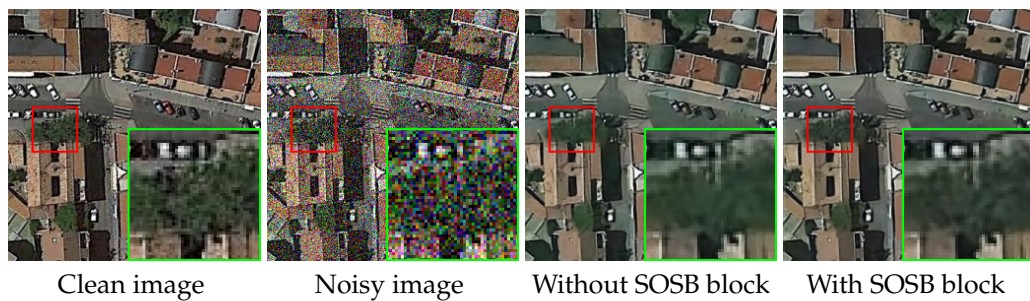

| Clean image | Noisy image | Without SOSB block | With SOSB block |

**Figure 21.** Enlarged detail comparisons of with/without SOSB block modules (with white Gaussian noise level $\sigma^2 = 50$).

**Table 10.** Comparison results of with/without SOSB modules. The bold values denote the best results.

| Metrics | Noisy Image | With SOSB Block | Without SOSB Block |
|---------|-------------|-----------------|--------------------|
| PSNR | 14.94 | **24.73** | 24.44 |
| SSIM | 0.4093 | **0.8143** | 0.7924 |

Overall, the ablation experiments illustrated that optimal choice of $\gamma_1$ and $\gamma_2$, along with the deployment of SOSB block modules, enabled our proposed AWTV$F^2$Net model to have better denoising ability while preserving the fine details of images.

*5.4. Experimental Results of Real-World Noisy Remote Sensing Images*

To further verify the ability of our proposed model for restoring real-world noisy remote sensing images, we used three real-world datasets: The AVRIS Indian Pines dataset [55], the ROSIS University of Pavia dataset [56], and the HYDICE Urban dataset [24]. As these real noisy remote sensing images do not include paired clean images, the full-reference image quality of PSNR and SSIM metrics are unsuitable for use in this subsection. Hence, the no-reference image quality measurement of the Spatial-Spectral Entropy-based Quality (SSEQ) [61], the Blind/Referenceless Image Spatial Quality Evaluator (BRISQUE) [62], and the blind image integrity notator using DCT statistics (BLIINDS-II) [63] were used to evaluate the quality of the denoised images. Using a bilinear operator, we resized all of real-world noisy images to $256 \times 256$. It is worth noting that our model was not proposed for blind image denoising. Therefore, we set the $\sigma^2 = 35$ for the AVRIS Indian Pines and the ROSIS University of Pavia datasets, and we used the model trained on a mixed of Gaussian and Speckle noise for denoising the HYDICE Urban dataset. The choices of noise level and the type of mixed noise were based on our rough estimation. The visual comparisons and numerical measurement comparisons are presented in Figure 22 and Table 11, respectively.

Figure 22 shows detailed comparisons for the three datasets. We present band 3 of the AVRIS Indian Pines dataset in the first row of Figure 22, while the second row is band 2 of the ROSIS University of Pavia Dataset. The last row is presented in band 205 of the HYDICE Urban dataset. From Figure 22, we can conclude that the DIPTV, DIPWTV, and DnCNN-B models obtained similar restored images in the AVRIS Indian Pines dataset, which were not much different from the noisy remote sensing image. Notably, the DIPTV and DIPWTV models could not restore images from the ROSIS University of Pavia dataset and the HYDICE Urban dataset. There was no content in the images obtained by the DIPTV and DIPWTV approaches. The DnCNN-B method could remove noise from these two datasets; however, it could not preserve the details of images. In contrast, our proposed AWTV$F^2$Net model not only removed the noise without losing detailed contents of images, but also made the images clearer than the original noisy images.

In Table 11, we present the mean SSEQ, BRISQUE, and BLIINDS-II metrics of the different methods on the three datasets. A higher SSEQ value denotes higher quality of the restored image, while a lower BRISQUE metric means that the quality of the denoised image is the higher. The BLIINDS-II metric is the same as the SSEQ: a higher value indicates higher quality of the restored image. As DIPTV and DIOWTV were invalid in the last two datasets, the mean values of SSEQ, BRISQUE, and BLIINDS-II could not be calculated (denoted by the '-'). From Table 11, it can be seen that our model performed best in the first and the last real noisy remote sensing image datasets. The DnCNN-B approach obtained the lowest value of BRISQUE on the ROSIS University of Pavia dataset. For the SSEQ and BLIINDS-II metrics, our model obtained the highest values in the same dataset. Overall, the proposed AWTV$F^2$Net presented superior performance over compared methods on the real noisy remote sensing image datasets.

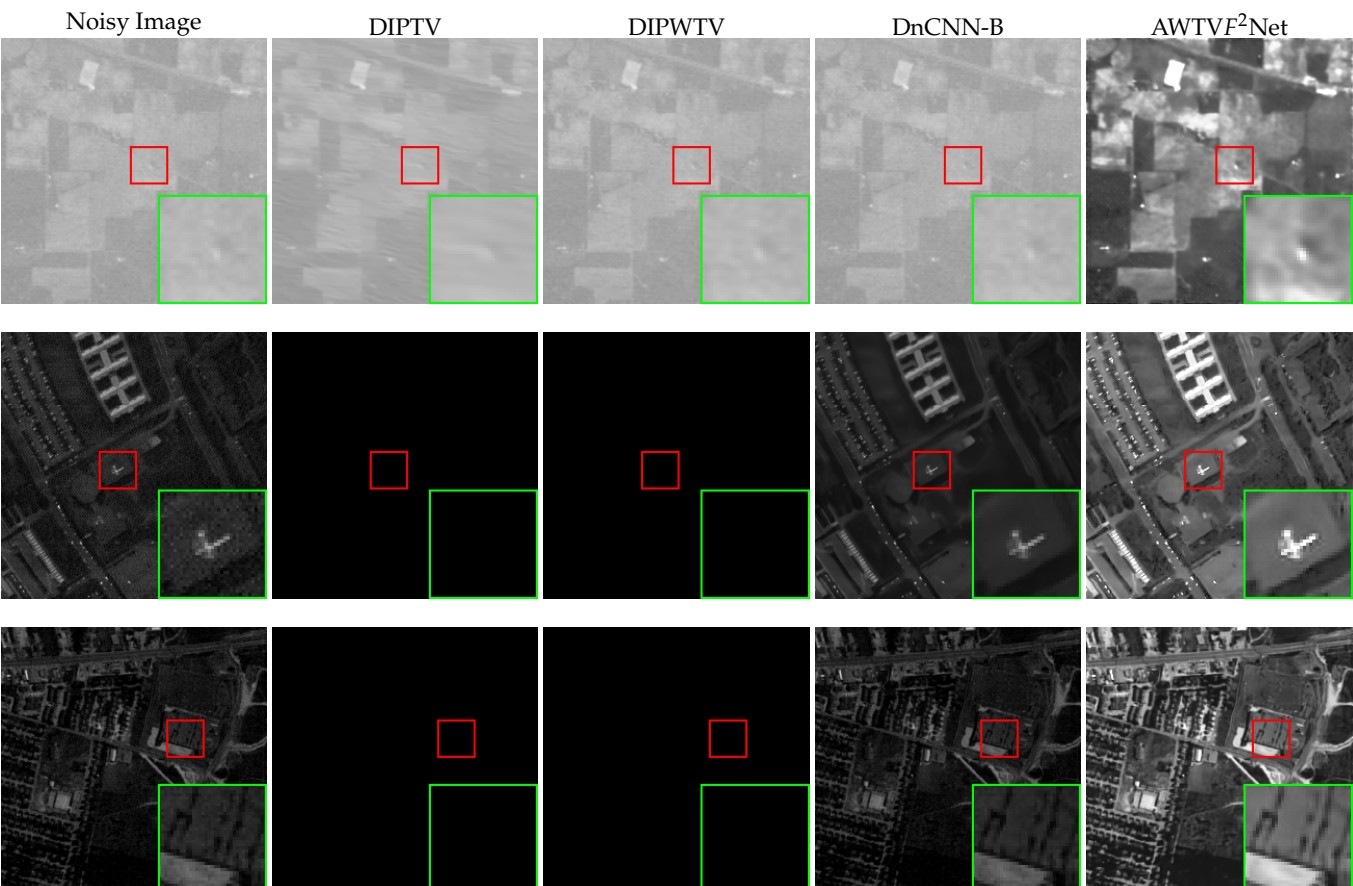

**Figure 22.** Restored details comparisons of real world noisy remote sensing image datasets.

**Table 11.** Numerical comparisons of different real noisy remote sensing image datasets. The bold values denote the best results.

| Dataset | Evaluation Method | DIPTV | DIPWTV | DnCNN-B | AWTV$F^2$Net |
|---------|-------------------|-------|--------|---------|-------------|
| AVIRIS Indian Pine dataset | SSEQ ↑ | 34.92 | 28.89 | 24.94 | **37.86** |
| | BRISQUE ↓ | 42.83 | 41.42 | 44.77 | **39.22** |
| | BLIINDS-II ↑ | **29.38** | 14.38 | 1.88 | **29.38** |
| ROSIS University of Pavia dataset | SSEQ ↑ | - | - | 21.30 | **26.45** |
| | BRISQUE ↓ | - | - | **25.85** | 26.82 |
| | BLIINDS-II ↑ | - | - | 21.75 | **30.08** |
| Urban | SSEQ ↑ | - | - | 24.28 | **35.27** |
| | BRISQUE ↓ | - | - | 34.01 | **30.67** |
| | BLIINDS-II ↑ | - | - | 15.50 | **26.83** |

From the experimental results in synthesized and real noisy remote sensing images denoising, the visual comparisons and numerical results well-demonstrated that our proposed AWTV$F^2$Net model can remove various types of noise effectively without losing the essential detail contents of remote sensing images. Moreover, the real-world noisy image experimental results indicate that our model is not only suitable for color image denoising, but also gray image restoration. The proposed method obtained superior performance over the other approaches used for comparison. The key merit of our proposed model is preventing fine feature information from smoothing in the denoising process.

## 6. Discussion

This study provided a framework for pre-processing noisy images in real-world remote sensing applications. We proposed an anisotropic weighted total variation feature fusion network consisting of four modules: AWTV-Net, SOSB, AuEncoder, and FB. The

denoising capability of the proposed model was evaluated through experiments on different benchmark and real-world datasets. By comparing the results of our method with the other methods, we confirmed that AWTV$F^2$Net could obtain a higher quality of denoised remote sensing images. Compared with the other methods, the proposed framework easily preserved more the fine details in the restored images. The AWTV-Net module of this study is a modified U-Net, which differs from the original U-Net [47]. The original U-Net integrates the skip layer outputs into the feature maps of the decoder, while our framework inherits the original U-Net and uses SOSB modules to boost the skip layer outputs. This combined use of the SOSB module and skip layers allows our framework to extract more critical detailed information about remote sensing images. Different from DIPWTV [22], the proposed weight function is an anisotropic diffusion coefficient function. The anisotropic diffusion coefficient can change as the algorithm iteration, thus adaptively preserving the small contents in restored images. This is a significant merit of the model proposed in this context over State-of-The-Art (SoTA) models, as it can obtain high-quality denoised sensing images while preserving small textures.

The excellent results obtained can be attributed to the following: (1) Multi-scale input sources are processed by the AuEncoder modules, enriching the feature space of reconstruction images; (2) multi-level feature maps are concatenated by FB modules to provide comprehensive essential information of remote sensing images, also as mentioned in [49]; and (3) the novel anisotropic diffusion coefficient function and suitable weights for loss functions allow the framework to stay robust under different datasets and noise cases.

We focused on the remote sensing image restoration task based on the proposed AWTV$F^2$Net. However, it is worth noting that there are some limitations when using the proposed framework: (1) our model is not designed for blind image denoising, therefore, before using our framework, it is recommended to (at least roughly) estimate the noise intensity [49]; (2) in the hyperspectral image denoising task, it is recommended to account for the lightness and image contrast [13], which may directly affect the restoration quality of remote sensing images; (3) for the mixed noise removal task, we mainly focused on the low level of noisy images, as the heavy mixed noise could cause loss of information in the restored images, also mentioned in [40]; and (4) our network was designed with many layers to preserve image texture information, which led to it not being applicable to mobile devices. Thus, potential further work includes designing a blind denoising network for noisy remote sensing images, exploring the relationship between the lightness and the quality of restored remote sensing images, and improving the contrast of denoised images with low lightness. Furthermore, we intend to investigate further the design of a lightweight version of our network to retain superior performance under constrained computation scenarios, including its application in mobile devices.

## 7. Conclusions

Remote sensing image restoration is a critical task in the remote sensing image processing field. We proposed a novel denoising deep learning network, named the anisotropic weighted total variation feature fusion network (AWTV$F^2$Net) framework, in this work, which provides an example of fusing extracted maps from the traditional model with feature maps from deep convolutional neural networks. Comprehensive experiments were conducted to test and validate the performance of our proposed AWTV$F^2$Net model. These experiments mainly included three aspects: removing the white Gaussian noise, removing mixed noise, and processing real noisy remote sensing images. First, we compared the effectiveness of our approach on the NWPU, PatternNet, and UCL datasets. For the Gaussian noise removal task with different noise levels on the three benchmark datasets, our model's PSNR and SSIM values were 0.12~7.73 dB and 0.0327~0.4750 higher, respectively, than other methods used for comparison. In the mixed noise removal task, our model's PSNR and SSIM values were 0.46~19.39 dB and 0.0237~0.5362 higher, respectively, on the three datasets. The SSEQ and BLIINDS-II, and BRISQUE values of AWTV$F^2$Net on the three real-world datasets were 3.94~12.92, 8.33~27.5 higher, and 2.2~5.55 lower than those of

the compared methods, respectively. The final restored images of our method can preserve more texture details than those of other compared algorithms. In addition, as the noise level increased, the proposed model still obtained superior performance over the other methods. Extensive experiments demonstrated that our model can achieve satisfactory results through objective numerical metrics, and it obtained high-quality denoised remote sensing images without the loss of detailed textures.

**Author Contributions:** Conceptualization, methodology, data analyses, original draft preparation and reviewing of the bibliography: H.Q. Methodology, data analyses, funding acquisition, manuscript review: S.T. Data analyses, funding acquisition, and manuscript review: Z.L. All authors have read and agreed to the published version of the manuscript.

**Funding:** This work was funded by the Informatization Plan of Chinese Academy of Sciences (Grant number: CAS-WX2021PY-0109), the National Key Research and Development Program of China (Grant No. 2018AAA0101001), the National Natural Science Foundation of China (Grants No. 11731004, No. 12126323 and No. 11761141005), the Shanghai Science and Technology Commission Foundation (Grants No. 22DZ2229014 and No. 20511100200).

**Data Availability Statement:** The original code alnd test datasets are available online at https://github.com/qihuiqing/AWTVFFNet, accessed on 1 November 2022.

**Conflicts of Interest:** The authors declare no conflict of interest.

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
