# Peer review of "Anisotropic Weighted Total Variation Feature Fusion Network for Remote Sensing Image Denoising"

_remotesensing, doi:10.3390/rs14246300_

Round 1

Reviewer 1 Report

This manuscript proposed a novel anisotropic weighted total variation features fusion network (AWTVF2Net) based on the deep learning approach for remote sensing image denoising, which has significantly superior performance in noise removal and preserving detailed contents of texture features. However, here are some comments to improve paper further better.

1. Line 1-20 describes the background, methods and results of the manuscript, but there is a lot of text, so it is recommended that author revise part of the content.

References:

[1] Han, L., Zhao, Y., Lv, H., Zhang, Y., Liu, H., & Bi, G. (2022). Remote Sensing Image Denoising Based on Deep and Shallow Feature Fusion and Attention Mechanism. Remote Sensing, 14(5), 1243.

[2] Wang, Z., Ng, M. K., Zhuang, L., Gao, L., & Zhang, B. (2022). Nonlocal Self-similarity-based Hyperspectral Remote Sensing Image Denoising with 3D Convolutional Neural Network. IEEE Transactions on Geoscience and Remote Sensing.

2. Line 24-52 introduced the significance of the study and the types of remote sensing image noise, but these three paragraphs have some repetitive content, resulting in a long space, which can be reorganized into one paragraph. Beside, it is recommended to focus on the current research on remote sensing image denoising.

3. Line75-101 summarized the application of deep learning and the method proposed in this paper. It is not recommended that the author use two paragraphs to describe.

4. Line148-190, when summarizing the traditional methods, the research results should be summarized and sorted out, so as to avoid a large number of who proposed what method, such as Xu et al. Proposed、Riya et al. Proposed and so on.

References:

[1] Wang, L., Ma, H., Li, J., Gao, Y., Fan, L., Yang, Z., ... & Wang, C. (2022). An automated extraction of small-and middle-sized rice fields under complex terrain based on SAR time series: A case study of Chongqing. Computers and Electronics in Agriculture, 200, 107232.

5. Line207-243 has the same problem as the fourth point.

6.The right of Figure 3 can be used to compare the diffusion coefficient function curve, so it is recommended that the author delete the left picture.

7.The name of Figure 4 is relatively brief, so it is recommended that author explain each module of the network in the figure.

8.Line355-384 introduced the datasets, and it is recommended that author describe them in sections, like 4.1.1 Benchmark datasets, 4.1.2 Real-World Datasets.

9. In section 5 “Results and Discussion”, the manuscript described the experimental results of additive white gaussian noise, mixture noise, ablation study and real-world noisy remote sensing images, but this part is long and affects the readability of the paper, so there are some comments:

(1) Generally, results and discussions belong to two separate contents. Therefore, it is recommended that author organize the experimental results into Section 4, and focus on sorting out the results of the the comparative experiments.

(2) In discussion, it is recommended that author focus on summarizing the differences between the proposed method and the other methods, convergence analysis, feature analysis and the computational efficiency.

References:

[1] Feng, X., Zhang, W., Su, X., & Xu, Z. (2021). Optical Remote sensing image denoising and super-resolution reconstructing using optimized generative network in wavelet transform domain. Remote Sensing, 13(9), 1858.

[2] Dou, H. X., Pan, X. M., Wang, C., Shen, H. Z., & Deng, L. J. (2022). Spatial and Spectral-Channel Attention Network for Denoising on Hyperspectral Remote Sensing Image. Remote Sensing, 14(14), 3338.

10.The English writing should be improved. It is better to edit it by native English speaker.

Author Response

This manuscript proposed a novel anisotropic weighted total variation features fusion network (AWTVNet) based on the deep learning approach for remote sensing image denoising, which has significantly superior performance in noise removal and preserving detailed contents of texture features. However, here are some comments to improve the paper further better.

Thank you very much for your comments and suggestions that allow us to greatly improve the quality of the manuscript. We have greatly improved our manuscript as follows. In addition, the revised manuscript has been checked by an English language native speaker using the MDPI author service.

  1. Line 1-20 describes the background, methods, and results of the manuscript, but there is a lot of text, so it is recommended that the author revise part of the content.

References:

[1] Han, L., Zhao, Y., Lv, H., Zhang, Y., Liu, H., & Bi, G. (2022). Remote Sensing Image Denoising Based on Deep and Shallow Feature Fusion and Attention Mechanism. Remote Sensing, 14(5), 1243.

[2] Wang, Z., Ng, M. K., Zhuang, L., Gao, L., & Zhang, B. (2022). Nonlocal Self-similarity-based Hyperspectral Remote Sensing Image Denoising with 3D Convolutional Neural Network. IEEE Transactions on Geoscience and Remote Sensing.

Response: We are grateful for your comments and valuable references. We have cited the proposed references in the Introduction section ([6][10]). We improved the text in lines 1-20 and added the quantitative results in the abstract (lines 8-14 ). The revised version of the abstract is as follows:

“Remote sensing images are widely applied in instance segmentation and objetive recognition; however, they offen suffer from noise, influencing the performance of subsequent applications. Previous image denoising works have only obtained restored images without preserving detailed texture. To address this issue, we proposed a novel model for remote sensing image denoising, called the anisotropic weighted total variation feature fusion network (AWTVNet), consist of four novel modules (WTV-Net, SOSB, AuEncoder, and FB). AWTVNet combines traditional total variation with a deep neural network, improving the denoising ability of the proposed approach. Our proposed method is evaluated by PSNR and SSIM metrics on three benchmark datasets (NWPU, PatternNet, UCL), and the experimental results show that AWTVNet can obtain  0.12~19.39dB/0.0237~0.5362 higher on PSNR/SSIM values in the Gaussian noise removal and mixed noise removal tasks than state-of-the-art algorithms. Meanwhile, our model can preserve more detailed  texture features. The SSEQ, BLIINDS-II, and BRISQUE values of AWTVNet on the three real-world datasets (AVRIS Indian Pines, ROSIS University of Pavia, HYDICE Urban) are 3.94~12.92 higher, 8.33~27.5 higher, and 2.2~5.55 lower than those of the compared methods, respectively. The proposed framework can guide subsequent remote sensing image applications, regarding the pre-processing of input images”

  1. Lines 24-52 introduced the significance of the study and the types of remote sensing image noise, but these three paragraphs have some repetitive content, resulting in a long space, which can be reorganized into one paragraph. Besides, it is recommended to focus on the current research on remote sensing image denoising.

Response: We have greatly improved the text in lines 24-52 according to your suggestions. The revised content is presented in lines 19-23. The suggested references were added in our introduction ([1], [6], [10], [11],[12]).

[1] Wang, L.; Ma, H.; Li, J.; Gao, Y.; Fan, L.; Yang, Z.; Yang, Y.; Wang, C. An automated extraction of small-and middle-sized rice 738 fields under complex terrain based on SAR time series: A case study of Chongqing. Computers and Electronics in Agriculture 2022, 200, 107232.

[6] Han, L.; Zhao, Y.; Lv, H.; Zhang, Y.; Liu, H.; Bi, G. Remote Sensing Image Denoising Based on Deep and Shallow Feature Fusion 749 and Attention Mechanism. Remote Sensing 2022, 14, 1243.

[10] Wang, Z.; Ng, M.K.; Zhuang, L.; Gao, L.; Zhang, B. Nonlocal Self-similarity-based Hyperspectral Remote Sensing Image Denoising with 3D Convolutional Neural Network. IEEE Transactions on Geoscience and Remote Sensing 2022.

[11] Feng, X.; Zhang, W.; Su, X.; Xu, Z. Optical Remote sensing image denoising and super-resolution reconstructing using optimized generative network in wavelet transform domain. Remote Sensing 2021, 13, 1858.

[12] Dou, H.X.; Pan, X.M.; Wang, C.; Shen, H.Z.; Deng, L.J. Spatial and Spectral-Channel Attention Network for Denoising on Hyperspectral Remote Sensing Image. Remote Sensing 2022, 14, 3338.

  1. Lines 75-101 summarized the application of deep learning and the method proposed in this paper. It is not recommended that the author use two paragraphs to describe.

Response: We reorganized the content in lines 24-38. The two paragraphs have been combined into one paragraph as follows:

“Researchers tend to use image processing methods to eliminate remote sensing noise signals, and traditional image denoising methods are usually limited by prior knowledge about the noise [8] and the parameters of these algorithms must be tuned [9], which is inevitable and complex. Recently, various studies have  proposed deep convolutional neural networks for image denoising. For example, Wang et al. [10] proposed a 3D convolutional neural network (CNN) for remote sensing image denoising; Feng et al. [11] utilized generative network (GN) technology to denoise remote sensing images; Dou et al. [12] used a spatial and spectral channel attention network for hyperspectral remote sensing image denoising. However, as remote sensing images contain rich texture information, previous deep neural networks could not allow the restored remote sensing images to preserve significant image textures efficiently. For this purpose, we proposed an anisotropic weighted total variation feature fusion network (AWTVNet) for remote sensing image denoising. The proposed network consists of an anisotropic weighted total variation (AWTV-Net) module and four strengthen--operate--subtract boosting strategy modules (SOSB), three auto-encoder modules (AuEncoder), and three fusion block modules (FB). The image denoising workflow of our model is shown in Figure 1.”

  1. Line148-190, when summarizing the traditional methods, the research results should be summarized and sorted out, so as to avoid a large number of who proposed what method, such as Xu et al. Proposed、Riya et al. Proposed and so on.

References:

[1] Wang, L., Ma, H., Li, J., Gao, Y., Fan, L., Yang, Z., ... & Wang, C. (2022). An automated extraction of small-and middle-sized rice fields under complex terrain based on SAR time series: A case study of Chongqing. Computers and Electronics in Agriculture, 200, 107232.

Response: We agree with you. We improved the text in lines 148-190 (see details in lines 83-116 in revised manuscript). We summarized the research works of filter-based methods lines 83-92. Lines 93-104 show the related works of the Total Variation models. Lines 105-116 described research works of Tensor decomposition-based algorithms. The revised version is as follows:

“Filter-based methods use the local information of the center pixel and remove noise according to the numerical relationship between the current pixel and neighboring pixels, such as mean filter [14], median filter [15], and Gaussian filter [16]. Filter-based methods tend to apply to image de-noising with a quick implementation property. However, filter-based methods provide the same coefficient in all directions, causing blurring in  fine features. Buades et al. [17] have proposed a non-local mean method to overcome the limitations of filter-based methods. This method can obtain higher-quality restored images than local information-based filters. The block matching and three-dimensional filtering (BM3D) algorithm [18] is more complicated than the non-local mean method, which combines the advantages of the non-local mean method and wavelet transform domain methods.

Total variation model [9] was proposed in 1992, which aims to preserve  image structure information. Total variation-based methods use regularization to constrain the denoising model, maintaining image texture features. Considering the spectral noise and spatial information difference in hyperspectral images, Yuan et al. [19] have proposed a spectral--spatial adaptive total variation (SSAHTV) model for hyperspectral remote sensing image denoising. Due to the  structural sparsity of hyperspectral images, a group sparsity regularized hybrid spatial--spectral total variation (GHSSTV) [20] model has been proposed for hyperspectral image restoration. In recent years, deep image prior [21, 22] information has been integrated into total variation models, which can make up for the unknown prior. However, the total variation-based methods require the tuning of complex parameters, which is usually not easy. Therefore, adaptive parameter selection for the weighted-TV model [23] has been used to address the complex parameters' tuning problem.

Tensor decomposition-based algorithms consider the inherent structure of remote sensing imagery, using low-rank tensor decomposition and recovery technology for hyperspectral image restoration. Wang et al. [24] have utilized the consistent structures between clean hyperspectral images and noisy images for image denoising by tensor decomposition and recovery. As the low-rank tensor approximation lost small content information, Zeng et al. [25] have added regularization to the low-rank tensor approximation item, which can improve the performance of remote sensing image restoration. Kong et al. [26] have proposed a framelet-tensor nuclear norm model for hyperspectral image denoising that takes full advantage of the redundancy of the framelet transform and the low-rank nature of the framelet-based transformed tensor. Overall, Tensor decomposition-based algorithms have excessively high computational and time costs, due to the need for decomposition on high-dimension tensors.”

  1. Line 207-243 has the same problem as the fourth point.

Response: We greatly improved our manuscript according to your comments. The revised contents are presented in lines 126-156. We summarized the related works of supervised deep learning methods in lines 129-143. Lines 144-156 described the unsupervised deep learning algorithms. The revised contents are as follows:

“Supervised deep learning methods for image denoising involve the use of paired clean and noisy images to train neural networks. The multi-layer perception (MLP) denoising network [27], consisting of four fully connected layers, was proposed by Burger et al. It was the first time that deep learning achieved similar performance to that of the BM3D algorithm in the image restoration task. Considering the high spectral correlation between adjacent bands in hyperspectral images, Maffei et al. [28] have utilized a single supervised model for hyperspectral image denoising, obtaining better results than BM3D and the MLP-based model. Combining the local and global information of noisy remote sensing images, a deep spatial--spectral global reasoning network based on the U-Net architecture was invented by Cao et al. [29] in 2021. This model used the U-Net network to extract rich features from the input images, in order to obtain high-quality denoised images. Jia et al. [30] have proposed a dual-complementary convolution network (DCCNet) for remote sensing image denoising. The DCCNet uses a wavelet transform operation and combines it with a shuffling operation to recover the image structure and texture information. Ulyanov et al. [31] have used a generator network to learn the prior from the random input noise vector, then restored the image from the prior information by a decoding network. All of the models mentioned above are supervised methods and require clean images for supervised learning during network training.

Unsupervised deep learning involves training neural networks with noisy images, and does not require clean images as in surpervised learning. Unsupervised deep learning methods for image denoising are also called blind denoising methods. As clean  remote sensing and medical images are hard to acquire, blind image denoising has become popular. There exist many blind denoising algorithms [32-37]. The "Noisy-As-Clean" (NAC) strategy [38] of training a self-supervised network for image denoising involves adding a simulated noise to the noisy image as input data and regarding the noisy images as target images. The NAC model further destroys the information of noisy images, leading to poor results. Huang et al. [39] have constructed paired images from the same noisy image using a random neighbor sub-sampler. This model avoids ruining the noisy image, but it lose information about the noisy image due to down-sampling, leading to unsatisfying results. A single-image capable speckling method for image denoising has been proposed by Wang et al. [40] in 2022, which presented better restoration results than previously mentioned unsupervised models.”

  1. The right of Figure 3 can be used to compare the diffusion coefficient function curve, so it is recommended that the author delete the left picture.

Response: We agreed with you. We deleted the left subplot in Figure 3 and improved the caption of the figure. Figure 3 mainly shows the comparisons of different diffusion coefficient functions with ours. Therefore, we revised the caption to “Comparisons of different diffusion coefficient function curves.“

  1. The name of Figure 4 is relatively brief, so it is recommended that the author explain each module of the network in the figure.

Response: We agree with you. We introduced the main modules of our framework in Section 3.2, including their functions and merits. According to your comments, we improved the caption of Figure 4 as follows:

Figure 4. Network architecture of the proposed AWTVNet. In this framework, the light green rectangles denote the middle features of the AWTV-Net module. The blue rectangle is the final output feature from AWTV-Net. The white rectangles are the skip feature maps. The thick, different-colored arrows indicate convolutional operators with different kernel sizes and strides, described in the middle dot-line rectangle. The black arrows denote the directions of data flowing. The SOSB rectangles are the feature map boosting modules, and AuEncoder rectangles are the modules extracting features from multi-inputs,   denotes additive operator. FB rectangles indicate the feature fusion modules. The red arrow marks the process of image reconstruction. The loss rectangles represent different loss calculations.”

  1. Line 355-384 introduced the datasets, and it is recommended that author describe them in sections, like 4.1.1 Benchmark datasets, 4.1.2 Real-World Datasets.

Response: We reorganized subsection 4.1. We have described the three benchmark datasets in subsection 4.1.1 and introduced the three real-world datasets in subsection 4.1.2, and shown in lines 296-326. The revised contents are as follows:

“4.1 Datasets

4.1.1 Benchmark Datasets

We used three benchmark datasets in our experiments to train and test AWTVNet. These three datasets were NWPU-RESISC45 (NWPU) [52], PatternNet [53], and UC Merced land-use dataset (UCL) [54]. Northwestern Polytechnical University created the NWPU-RESISC45 dataset for remote sensing image scene classification.The NWPU-RESISC45 dataset includes 45 classes, where each class has 700 images with a size of 256x256. All of these images are colorful, with a spatial resolution from 0.2--30m for most scene classes. The PatternNet dataset contains 38 classes, and each of which has 800 images. All remote sensing images in PatternNet have a spatial resolution from 0.062--4.693m with a size of 256x256 in the RGB color space. The UC Merced land-use dataset is a freely and publicly available remote sensing image dataset, composed of 21 classes with a total number of 2100 images. The images in the UC Merced land-use dataset have a spatial resolution of 0.3m and a size of 256x256 in the RGB color space.

4.1.2 Real-World Datasets

We compared the performance between our model and other methods on three real-world datasets: AVRIS Indian Pines dataset [55], the ROSIS University of Pavia dataset [56], and the HYDICE Urban dataset [57]. The AVIRIS Indian Pines dataset was gathered by an AVIRIS sensor over the Indian Pines test site in North-western Indiana, and consists of 145x145 pixels images with 224 spectral reflectance bands in the wavelength range of 0.4~2.5xmeters. The AVIRIS Indian Pines dataset includes some bands affected by the mixture of Gaussian and impulse noise, as introduced in [55]. For this study, we have used bands 3~6 to test our model. The ROSIS University of Pavia dataset is a hyperspectral image dataset gathered by a ROSIS sensor over Pavia, Italy. The size of images is 610x340 with 103 spectral bands. The geometric resolution of the images in the ROSIS University of Pavia dataset is 1.3 m. We used the first six bands' images in our experiments. The HYDICE Urban dataset is one of the most widely used hyperspectral remote sensing image datasets. The size of these images is 307x307, where each pixel corresponds to a 2x2  area. There are 210 wavelengths ranging from 400 nm to 2500 nm, resulting in a spectral resolution of 10 nm. We have chose bands 138, 203, and 205 for  our experiments.”

  1. In section 5 “Results and Discussion”, the manuscript described the experimental results of additive white gaussian noise, mixture noise, ablation study and real-world noisy remote sensing images, but this part is long and affects the readability of the paper, so there are some comments:

(1) Generally, results and discussions belong to two separate contents. Therefore, it is recommended that author organize the experimental results into Section 4, and focus on sorting out the results of the the comparative experiments.

(2) In discussion, it is recommended that author focus on summarizing the differences between the proposed method and the other methods, convergence analysis, feature analysis and the computational efficiency.

References:

[1] Feng, X., Zhang, W., Su, X., & Xu, Z. (2021). Optical Remote sensing image denoising and super-resolution reconstructing using optimized generative network in wavelet transform domain. Remote Sensing, 13(9), 1858.

[2] Dou, H. X., Pan, X. M., Wang, C., Shen, H. Z., & Deng, L. J. (2022). Spatial and Spectral-Channel Attention Network for Denoising on Hyperspectral Remote Sensing Image. Remote Sensing, 14(14), 3338.

Response: We agreed with you. We reorganized Section 5. We separated “Results and Discussion” into two sections: Section 5. Results and Section 6. Discussion.

Section 5 summarizes all the compared results of the three benchmark datasets, the three real-world datasets, and the ablation experiment.

Section 6 describes the limitations of this study and the difference between the proposed framework and the other methods:

“This study provided a framework for pre-processing noisy images in real-world remote sensing applications. We proposed an anisotropic weighted total variation feature fusion network consisting of four modules: AWTV-Net, SOSB, AuEncoder, and FB. The denoising capability of the proposed model was evaluated through experiments on different benchmark and real-world datasets. By comparing the results of our method with the other methods, we confirmed that AWTVNet could obtain a higher quality of denoised remote sensing images. Compared with the other methods, the proposed framework easily preserved more the fine details in the restored images. The AWTV-Net module of this study is a modified U-Net, which differs from the original U-Net [47]. The original U-Net integrates the skip layer outputs into the feature maps of the decoder, while our framework inherits the original U-Net and uses SOSB modules to boost the skip layer outputs. This combined use of the SOSB module and skip layers allows our framework to extract more critical detailed information about remote sensing images. Different from DIPWTV [62], the proposed weight function is an anisotropic diffusion coefficient function. The anisotropic diffusion coefficient can change as the algorithm iteration, thus adaptively preserving the small contents in restored images. This is a significant merit of the model proposed in this context over State-of-The-Art (SoTA) models, as it can obtain high-quality denoised sensing images while preserving small textures.

The excellent results obatined can be attributed to the following: (1) Multi-scale input sources are processed by the AuEncoder modules, enriching the feature space of reconstruction images; (2) multi-level feature maps are concatenated by FB modules to provide comprehensive essential information of remote sensing images, also as mentioned in [49]; and (3) the novel anisotropic diffusion coefficient function and suitable weights for loss functions allow the framework to stay robust under different datasets and noise cases.

We focused on the remote sensing image restoration task based on the proposed AWTVNet. However, it is worth noting that there are some limitations when using the proposed framework: (1) our model is not designed for blind image denoising, therefore, before using our framework, it is recommended to (at least roughly) estimate the noise intensity [49]; (2) in the hyperspectral image denoising task, it is recommended to account for the lightness and image contrast [13], which may directly affect the restoration quality of remote sensing images;  (3) for the mixed noise removal task, we mainly focused on the low level of noisy images, as the heavy mixed noise could cause loss of information in the restored images, also mentioned in [40]; and (4) our network was designed with many layers to preserve image texture information, which led to it could not be applied to mobile devices. Thus, potential further work includes designing a blind denoising network for noisy remote sensing images, exploring the relationship between the lightness and the quality of restored remote sensing images, and improving the contrast of denoised images with low lightness. Furthermore, we intend to investigate further the design of a lightweight version of our network to retain superior performance under constrained computation scenarios, including its application in mobile devices.”

  1. The English writing should be improved. It is better to edit it by native English speaker.

Response: In the process of revision, the manuscript was revised by all co-authors. Moreover, the revised manuscript has been checked by an English language native speaker using the MDPI author service.

Reviewer 2 Report

This research presents a new deep neural network to denoise remote sensing imagery. The related traditional work, novelty, theoretical equation, and PyTorch implementation are all clearly presented. The validations are comprehensive on different datasets and different noise types, including complexly mixed scenarios.

The article should be accepted for publication as is.

The only concern is the GitHub repo listed cannot be found, nor can I find it by search. I am very interested in trying the model myself.

Author Response

This research presents a new deep neural network to denoise remote sensing imagery. The related traditional work, novelty, theoretical equation, and PyTorch implementation are all clearly presented. The validations are comprehensive on different datasets and different noise types, including complexly mixed scenarios.

The article should be accepted for publication as is.

The only concern is the GitHub repo listed cannot be found, nor can I find it by search. I am very interested in trying the model myself.

Response: We are grateful for your comments. We reuploaded the code to the GitHub repo; the link is https://github.com/qihuiqing/AWTVFFNet. The code is currently public to everyone.

Reviewer 3 Report

A novel anisotropic weighted total variation features fusion network (AWTVF 2Net) based on the deep learning approach was proposed for Remote sensing image denoising. But paper needs an extensive changes prior to the publication process.

1.       Reduce the abstract and also add some quantitative findings in the abstract.

2.       Reduce the introduction section by reducing the irrelevant contents.

3.       Add more details in the section 2.3.

4.       Authors should clearly define the existing equations by adding suitable citation to them. Difficult to understand novel or modified equations proposed by the authors.

5.       Explain the equation 9 in better fashion, try to present suitable derivation.

6.       Why LeakyReLU was used?

7.       Conclusion should be reduced. Also, add some quantitative findings in the conclusion along with the suitable future work.

Author Response

A novel anisotropic weighted total variation features fusion network (AWTVNet) based on the deep learning approach was proposed for Remote sensing image denoising. But paper needs extensive changes prior to the publication process.

Thank you for your comments and suggestions. We have improved our manuscript according to the following comments. The revised manuscript has been checked by an English language native speaker using the MDPI author service.

  1. Reduce the abstract and also add some quantitative findings in the abstract.

Response: Thank you for your suggestions. We improved the abstract and added the quantitative results (lines 8-14). The revised abstract is as follows:

“Remote sensing images are widely applied in instance segmentation and objetive recognition; however, they offen suffer from noise, influencing the performance of subsequent applications. Previous image denoising works have only obtained restored images without preserving detailed texture. To address this issue, we proposed a novel model for remote sensing image denoising, called the anisotropic weighted total variation feature fusion network (AWTVNet), consist of four novel modules (WTV-Net, SOSB, AuEncoder, and FB). AWTVNet combines traditional total variation with a deep neural network, improving the denoising ability of the proposed approach. Our proposed method is evaluated by PSNR and SSIM metrics on three benchmark datasets (NWPU, PatternNet, UCL), and the experimental results show that AWTVNet can obtain  0.12~19.39dB/0.0237~0.5362 higher on PSNR/SSIM values in the Gaussian noise removal and mixed noise removal tasks than state-of-the-art algorithms. Meanwhile, our model can preserve more detailed  texture features. The SSEQ, BLIINDS-II, and BRISQUE values of AWTVNet on the three real-world datasets (AVRIS Indian Pines, ROSIS University of Pavia, HYDICE Urban) are 3.94~12.92 higher, 8.33~27.5 higher, and 2.2~5.55 lower than those of the compared methods, respectively. The proposed framework can guide subsequent remote sensing image applications, regarding the pre-processing of input images”

  1. Reduce the introduction section by reducing the irrelevant contents.

Response: We are grateful for your comments. We are sorry that there is redundant content in the Introduction section. We greatly improved the introduction by reducing the irrelevant content (lines 19-76).

We reduced the irrelevant contents: including the producing mechanism of noise, decreasing noise by improving imaging devices, and the repeat contents for introducing our proposed modules described in Section 3.2. Thus, lines 19-23 show the background and the types of noise. Lines 24-38 describe the limitations of the existing traditional methods and deep learning networks for remote sensing image denoising. Lines 39-67 are our main contributions in this study. Lines 68-76 are outlines of the remainder of this article.

In section 1, we described starting from the background of remote sensing image denoising to the types of noise, and then, we introduced the existing methods and their limitations; finally, we presented the main contributions of our works; reorganized in this way, it is easy to understand the contents of this section.

  1. Add more details in the section 2.3.

Response: Thank you for your suggestions. The revised contents of Section 2.3 are presented in lines 166-204 of our manuscript. In Section 2.3.1, we started to describe the diffusion model in image denoising. And then, we introduced the diffusion coefficient from diffusion models. Meanwhile, we showed three essential natures of diffusion coefficient functions. Using  coefficient function as an example shows the process of a diffusion coefficient function derived from the “tansig” function. Section 2.3.2 has described the total variation model. We introduced the discrete form of the regularization term and the value range of the “p” Norm. It is worth noting that many total variation models can be solved by the ADMM framework, which is one of the mature tools for solving non-convex problems.

We illustrated the added details of section 2.3 as follows. We added the description of the diffusion coefficient model in lines 166-167, which can lead to the diffusion coefficient function and its mathematical formula. Lines 176-179 show that we derive the existing coefficient function [43] from tansig function so that we can export our coefficient in the same way. In lines 180-188, we added descriptions of Figure 2. it can help to understand the properties of the diffusion coefficient function. In lines 194-204, we added illustrations on the total variation model and its solution method, which helped lead to our model in section 3.

  1. Authors should clearly define the existing equations by adding suitable citation to them. Difficult to understand novel or modified equations proposed by the authors.

Response: Thank you for your comments, and we are sorry for confusing this research novelty. We have added suitable citations to distinguish between the novel or modified equations and the existing ones in lines 206-226. The involved equations are mainly in lines 220-291 of Section 3. To introduce clearly and understand our work's novelty easily, we describe the involved equations (16)-(35) in the following text.

Equation (17) is our proposed model, and it is simplified into equation (18), then we get our model's simple version, equation (19). We solved model (19) by using the ADMM framework, which is a mature tool for solving non-convex problems. So the formula (20) is the augmented lagrangian function of our model. The solution of equation (20) is equivalent to finding the solutions of three sub-problems, which are equations (21)-(23). The anisotropic weight of our model is from the proposed diffusion coefficient, and the weight is changed as algorithm iterations, presented in equation (24). The formulas (27) - (32) describe the multi-level feature fusion of our framework from a mathematical point of view. Equation (33) is the proposed model's total variation loss, and (34) is the MSE loss of our network. Formula (35) is the weighted sum of equations (33) and (34) in this study.

In summary,

  • Equations (25)-(26) are cited from the study [48] (lines 245-248), involving the strengthen operate subtract boosting strategy;
  • The proposed novel or modified equations include equations (16)-(24) and (27)-(35). Equations (16)-(24) show the derivation process of our model and its solution, which are newly proposed in this study, presented in lines 206-226. Equations (27)-(32) show the process of multi-level feature fusion in our network; those equations are also newly proposed by us and shown in lines 283-288. Equations (33)-(35) show the loss functions of the proposed model, shown in lines 289-291.
  1. Explain equation 9 in a better fashion, and try to present a suitable derivation.

Response: Thank you for your suggestions. We are sorry for the unclear explanation of our proposed anisotropic diffusion coefficient function. We have greatly improved the content of equation (9), and now it is equation (12) in the revised manuscript. We derived our coefficient function from the “tansig” function and compared it with the existing diffusion coefficients. We have demonstrated our proposed coefficient function is the lower bound of  by mathematical theory and numerical simulation. To explain equation (12) better, we have shown some derivations, which involve equations (11)-(15). The revised contents are presented in lines 206-219 of our manuscript.

  1. Why LeakyReLU was used?

Response: LeakyReLU is a variant of ReLU. Compared with other activation functions such as Sigmoid and Tanh, ReLU converges faster and obtains faster calculation speed. However, ReLU causes a "Dead Neuron" problem. When the input value of ReLU is negative, its output value is zero, which makes the corresponding neural network nodes' gradients zero. Therefore, the parameters of dead neural network nodes can not be updated in this situation. LeakyReLU inherits the advantages of ReLU and makes up for the shortcomings of ReLU. When the input value of LeakyReLU is negative, its output value is leaky. The corresponding neural network nodes' gradients are not equal to zero, and then the parameters can be updated. In our study, we want to keep more details in restored remote sensing images. All neural network nodes of our proposed framework should be updated in every iteration. So, we use the LeakyReLU function activation function.

  1. Conclusion should be reduced. Also, add some quantitative findings in the conclusion along with the suitable future work.

Response: To present a clear conclusion, we have greatly improved the conclusion of this study. We added quantitative metrics for evaluating the proposed model on the three benchmarks and real-world datasets to illustrate the better performance of our model than that of the compared methods. First, we compared the effectiveness of our approach on the NWPU, PatternNet, and UCL datasets. For the Gaussian noise removal task with different noise levels on the three benchmark datasets, our model's PSNR and SSIM values were 0.12~7.73dB and 0.0327~0.4750 higher, respectively, than other methods used for comparison. In the mixed noise removal task, our model's PSNR and SSIM values were 0.46~19.39dB and 0.0237~0.5362 higher, respectively, on the three datasets.The SSEQ and BLIINDS-II, and BRISQUE values of AWTVF^2Net on the three real-world datasets were 3.94~12.92, 8.33~27.5 higher, and 2.2~5.55 lower than those of the compared methods, respectively. These added quantitative findings are presented in lines 710-718. The revised conclusion is as follows:

“Remote sensing image restoration is a critical task in the remote sensing image processing field. We proposed a novel denoising deep learning network, named the anisotropic weighted total variation feature fusion network (AWTVNet) framework, in this work, which provides an example of fusing extracted maps from the traditional model with feature maps from deep convolutional neural networks. Comprehensive experiments were conducted to test and validate the performance of our proposed AWTVNet model. These experiments mainly included three aspects: removing the white Gaussian noise, removing mixed noise, and processing real noisy remote sensing images. First, we compared the effectiveness of our approach on the NWPU,  PatternNet, and UCL datasets. For the Gaussian noise removal task with different noise levels on the three benchmark datasets, our model's PSNR and SSIM values were 0.12~7.73dB and 0.0327~0.4750 higher, respectively, than other methods used for comparison. In the mixed noise removal task, our model's PSNR and SSIM values were 0.46~19.39dB and 0.0237~0.5362 higher, respectively, on the three datasets.The SSEQ and BLIINDS-II, and BRISQUE values of AWTVNet on the three real-world datasets were 3.94~12.92, 8.33~27.5 higher, and 2.2~5.55 lower than those of the compared methods, respectively. The final restored images of our method can preserve more texture details than those of other compared algorithms. In addition, as the noise level increased, the proposed model still obtained superior performance over the other methods. Extensive experiments demonstrated that our model can achieve satisfactory results through objective numerical metrics, and it obtained high-quality denoised remote sensing images without the loss of detailed textures.”

Round 2

Reviewer 1 Report

  This research presents a new deep learning approach for Remote sensing image denoising, and the related theoretical methods, results and novelties are clearly described.In addition, the author publishes the code of the model, which increases readers' interest in reading and comprehension of the proposed model.

  The article should be accepted for publication.

Reviewer 3 Report

Authors have revised all my suggestions.